# Aviation passenger carbon footprint calculator with comprehensive emissions, life cycle coverage, and historical adjustment
Finn McFall [1] ✉, Dabo Guan [2,3], Jonathan Chenoweth [1], Xavier Font [4], Ionut Corduneanu[5], Eduard Goean [4,5] & Jhuma Sadhukhan [1]

Passenger aviation carbon footprint calculators often lack breadth, accuracy, transparency, and communication effectiveness, leading to underestimations of environmental impact and mistrust. This study addresses these gaps by developing a comprehensive methodology that broadens scope and improves accuracy. It incorporates nitrogen oxides, water vapour, contrail-induced cloudiness, upstream emissions from in-flight services, and life cycle emissions from aircraft and airports, offering a complete carbon footprint assessment. Accuracy is improved through detailed modelling of flight distance, fuel consumption, and emissions allocation adjusted for passenger class, luggage, and cargo. Historical adjustment factors refine pre-flight estimates by integrating real-world variations. The tool outputs a full emissions breakdown by source, offering unparalleled granularity and clarity. Validated against over 30,000 historical flights, the historical adjustment factor model achieves ~0.5% mean squared percentage error and shows current methods underestimate emissions. This study sets a standard for aviation carbon footprint calculators by enabling transparent, dynamic assessments for industry stakeholders.

In recent years, the aviation sector has seen immense growth, with the number of flights and passengers reaching all-time highs before the impact of the global COVID-19 pandemic. Prior to the pandemic, the aviation sector accounted for ~2.4% of annual human-caused $CO_2$ emissions, and it has been responsible for ~4% of the observed human-induced global warming to date[1]. Airbus forecasts that demand for passenger traffic will grow annually by 3.6% over the next 20 years, with projections indicating it could contribute to a 0.1 °C increase in global temperatures by 2050[1]. The aviation industry's striking contribution to greenhouse gas (GHG) emissions underscores the urgent need for accurate and reliable methods to quantify the carbon footprint of commercial aviation. Addressing this challenge directly supports progress towards multiple United Nations Sustainable Development Goals (SDGs), including Climate Action (SDG 13), Responsible Consumption and Production (SDG 12), and Industry, Innovation, and Infrastructure (SDG 9)[2].

Current carbon calculators of the impact of aviation fall short of what is required on several accounts. Their narrow scope means that they fail to capture the full breadth and depth of emission sources[3–5]. The lack of accuracy in each tool results from shortcomings in capturing user-specific, reputable, up-to-date and consistent data, and the lack of consistency between tools comes from not deploying standardised methodologies[4–9]. These shortcomings result in inconsistent and inaccurate assessments, posing a serious challenge for airlines, policymakers, and passengers, who rely on precise data to make informed decisions aimed at mitigating environmental impacts[10–12].

One critical shortcoming of many existing carbon calculations is the lack of inclusion of the non-Kyoto impacts, a term we use to refer to substances created by aeroplanes not covered under the Kyoto Protocol, such as nitrogen oxide ($NO_x$), water vapour ($H_2O$), and contrail-induced cloudiness (CiC), that contribute to aviation's impact on climate[13]. The non-Kyoto effects of aviation on climate can be substantially greater than the $CO_2$ effects[13], and therefore omitting these effects from carbon calculators results in a severe underestimation of the climate impact of a flight[14,15]. The Intergovernmental Panel for Climate Change suggests using a radiative

[1]Centre for Environment and Sustainability, University of Surrey, Guildford, Surrey, UK. [2]The Bartlett School of Construction and Project Management, University College London, London, UK. [3]Department of Earth System Sciences, Tsinghua University, Beijing, China. [4]Surrey Business School, University of Surrey, Guildford, Surrey, UK. [5]Therme Group RHTG AG, Vienna, Austria. ✉e-mail: f.mcfall@surrey.ac.uk

forcing index (RFI) multiplier, a measure of the importance of aircraft-induced climate change other than that from the release of fossil carbon alone. Effective radiative forcing is a measure of the rapid response of the climate system to external factors[16]. Contrail cirrus cloud formation produces the largest positive effective radiative forcing, followed by $CO_2$ and $NO_x$, while sulphate aerosols produce cooling[13]. Lee et al.[13] estimated that the Global Warming Potential over a 20-year time frame ($GWP_{20}$) for contrail cirrus compared to the $CO_2$ impacts (the $CO_2e/CO_2$ ratio) to be 2.3 and over 100 years ($GWP_{100}$) to be 0.63; they estimated the total non-$CO_2$ impacts of aviation compared to the $CO_2$ impacts to be 4.0 ($GWP_{20}$) and 1.7 ($GWP_{100}$). Teoh et al.[17], however, estimated the $GWP_{20}$ of contrail cirrus to be 1.1 and the $GWP_{100}$ to be 0.29. They suggest that the difference between their estimates and those of Lee et al.[13] is due to their analysis incorporating more air traffic in the sub-tropics, where persistent contrails are less likely to form. The non-$CO_2$ impacts of aviation on climate are very significant[13] and at least twice as large as the $CO_2$ impacts, with large uncertainties. Including contrail-induced emissions in the carbon footprint calculation acknowledges both a challenge and an opportunity. While contrails may not form for every flight, the modularity of the Air Travel Passenger Dynamic Emissions Calculator (ATP-DEC) allows for flexible modelling based on stakeholder choice. ATP-DEC's dynamic, parameterised model offers a more scientifically robust and comprehensive assessment than simplified methods such as RFI and contrail "bucket" classification[18]. The bucket approach provides high-level risk categories based solely on contrail patterns. It omits major contributors like $NO_x$ and water vapour and lacks sensitivity to flight-specific parameters. By failing to convey actual climate impact, it offers no transparent quantification and risks misleading users into underestimating the true scope of non-Kyoto aviation effects.

Similarly, very few calculators adopt a full life cycle assessment (LCA) perspective. A cradle-to-grave LCA approach must include the production and transportation of fuel (Well-To-Tank (WTT)), fuel burning during flying, take-off and landing (Tank-To-Wake (TTW)), in-flight services, and airport and aircraft life cycles[19,20]. Following an evaluation of existing carbon calculator tools, this paper outlines a comprehensive LCA-based methodology for the ATP-DEC, a novel calculator that improves the accuracy of the variables commonly used and the scope of those variables either ignored or poorly estimated. Compared to current tools in the market, ATP-DEC i) better raises awareness of the impact of flying and attributes such impact to different sources, ii) more accurately tracks and reports impacts for corporate reporting, iii) feeds more accurate data to carbon offsetting mechanisms[21], iv) identifies specific aspects of $CO_2e$ that change over time, v) better informs policies such as aviation taxation. By addressing the limitations of existing methods, this research aims to provide a robust framework that can be easily adopted by stakeholders within the aviation industry. The methodology is analysed over 30,000 flights, and all techniques and variables are explained and justified. Finally, the limitations of the model are stated for complete transparency[21].

## Results
### Comparison of ATP-DEC with existing tools
To benchmark the proposed methodology, carbon footprint estimates were compared against four existing calculators: Google Travel Impact Model (TIM), IATA $CO_2$ Connect, ICAO Carbon Emissions Calculator, and MyClimate. The selection of comparison tools was informed by their prevalence in industry practice and public use, reflecting their relative popularity and accessibility. These tools represent a diverse set of methodologies and serve as benchmarks for assessing the methodological advancements offered by ATP-DEC. The estimations for each of the tools are taken from their online calculator websites, with variables kept the same for each calculator where possible. Results are presented in Fig. 1, which visualises the carbon footprint of a flight between Singapore Changi (SIN) and Zurich (ZRH) across four passenger classes—First, Business, Premium Economy, and Economy—using stacked bars for ATP-DEC and single bars for the existing tools.

Figure 1 highlights a key distinction of the ATP-DEC: the ability to break down emissions into multiple sources for each passenger class: TTW, WTT, carry on and checked luggage, in-flight services, airport and aircraft life cycle emissions, non-Kyoto gases: $NO_x$, $H_2O$ and non-Kyoto impact: CiC. In contrast, the results from TIM, IATA, ICAO, and MyClimate are presented as aggregated emissions, limiting their ability to capture detailed contributors to the carbon footprint. A full breakdown of emissions not only allows stakeholders to decide which emissions matter for their requirements but also facilitates targeted environmental action by highlighting key areas for intervention.

ATP-DEC consistently produces higher overall emission estimates compared to existing tools, particularly for premium passenger classes (First and Business). This discrepancy is primarily due to the inclusion of non-Kyoto impacts ($NO_x$, CiC, $H_2O$), which are excluded by TIM, IATA, and ICAO. While MyClimate includes non-$CO_2$ effects using a simplified RFI, ATP-DEC's dynamic modelling of $NO_x$, $H_2O$, and CiC more accurately reflects their contributions[22] (Equation S1–S35). TIM 3.0 introduces contrail warming "buckets" with the emissions footprint, whereby contrail warming impact is categorised into relative impact levels compared to the fuel burn emissions[18]. A quantitative estimate of the actual climate impact is not provided, making it impossible to compare or aggregate non-$CO_2$ effects in a transparent way.

Additionally, the Historical Adjustment Factors (HAFs) derived from historical flight data further enhance precision by accounting for actual flight deviations and rerouting (Equation S19), which are not reflected in the static assumptions of the other tools. In this case, this real-world adaptability has captured the increased distance and change in mean latitude, increasing the fuel burn and non-Kyoto impacts.

When non-Kyoto impacts are excluded, the ATP-DEC results align more closely with TIM and IATA. However, ATP-DEC still demonstrates superior accuracy due to improved seat class weightings (Equation S10). By incorporating airline and aircraft-specific seating data (acquired from SeatGuru[23]), ATP-DEC produces a more realistic split of emissions between passenger classes. This contrasts with the generalised weightings used by other tools, which fail to account for variations in aircraft configurations and premium seating occupancy.

### HAF impact comparison
To evaluate the impact of HAFs on carbon footprint estimates, ATP-DEC was compared against post-flight emissions estimates derived from exact flight tracking data. A TIM-based method is implemented in two scenarios for comparison. This analysis was performed for flights from London Heathrow (LHR) and Incheon (ICN) over the course of 2023, as shown in. Again, input parameters have been kept constant apart from the flight route.

The analysis compares four scenarios:
- ATP-DEC using HAF (30-day rolling window) (green solid line): adjusts emissions dynamically using historical data trends from the preceding 30 days, accounting for operational deviations. This is a realistic scenario showing how the ATP-DEC operates.
- Theoretical ATP-DEC using actual post-flight distance (blue dashed line): estimates emissions using exact post-flight positional flight data. This is unrealistic in practice, but useful for the validation of HAF.
- TIM-based method scenario (red solid line): simulates a normal TIM-based method scenario (with constant distance). This scenario shows how the TIM-based method operates in reality.
- Theoretical TIM-based method using actual post-flight distance (orange dashed line): simulates a theoretical TIM-based method scenario by incorporating the actual flight distance to compare with the normal TIM-based method. This is an unrealistic scenario, but useful for validation.

Figure 2 demonstrates that the HAFs allow the output to closely match the emissions trend derived from post-flight data throughout the year. This result highlights the effectiveness of the rolling HAF in capturing real-world

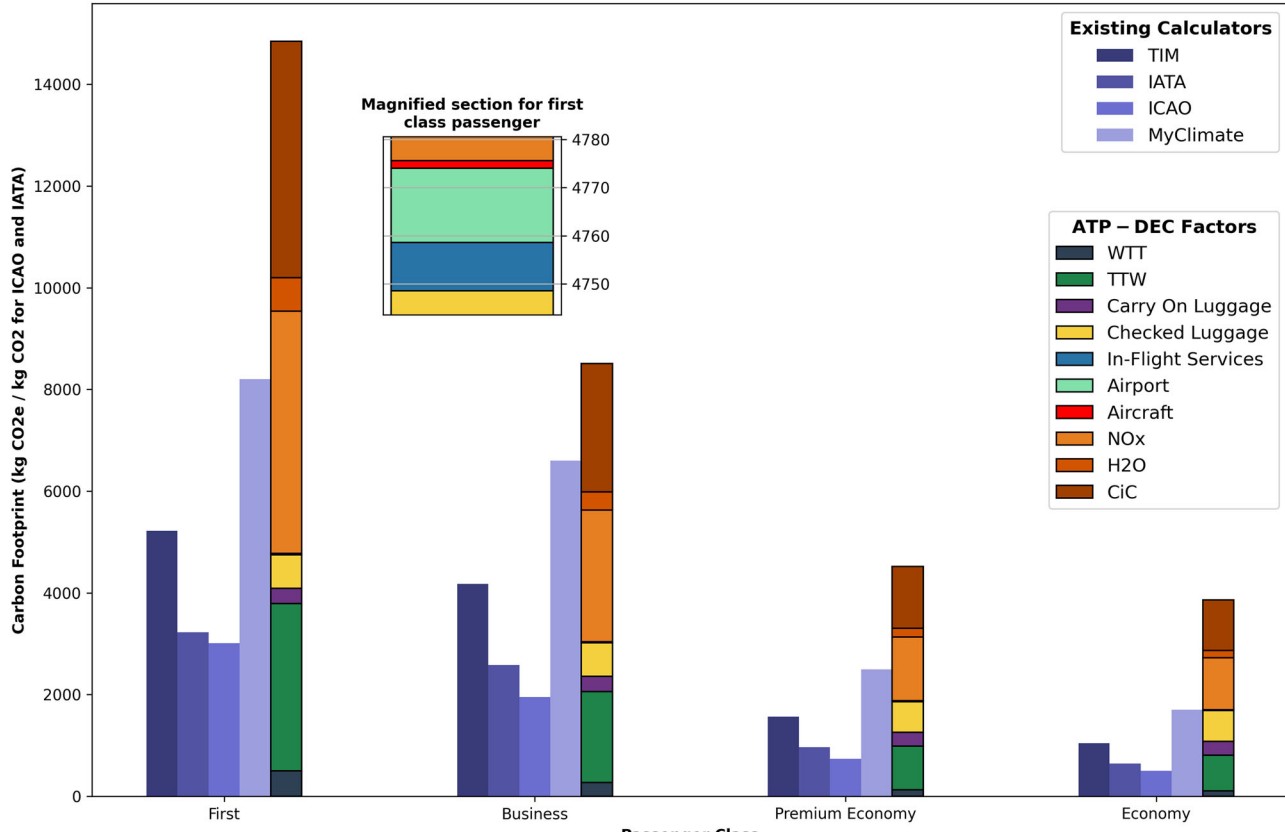

**Fig. 1 | Comparison of air passenger emission footprint calculator outputs by travel class for a flight from SIN to ZRH[30].** Bar chart showing the emissions footprint (in kg $CO_2$e for ATP-DEC, TIM, and MyClimate and kg $CO_2$ for IATA and ICAO) for a flight from Singapore (SIN) to Zurich (ZRH) across four passenger classes: first, business, premium economy, and economy. Four blue shades represent existing calculators, from dark to light: TIM, IATA, ICAO, and MyClimate. Stacked coloured bars represent the ATP-DEC results, with individual emissions components: Well-To-Tank (WTT, dark blue), Tank-To-Wake (TTW, green), carry-on luggage (purple), checked luggage (yellow), in-flight services (light blue), airport (teal), aircraft (red), nitrogen oxides ($NO_x$, orange), water vapour ($H_2O$, light brown), and contrail-induced cloudiness (CiC, dark brown). The inset magnifies the First Class ATP-DEC for improved visibility of small components.

variations, such as route deviations, air traffic inefficiencies, and operational changes, without relying on exact post-flight inputs. In this scenario, the flight path is altered due to the closure of Russian airspace, causing an increase in distance, fuel consumption, and emissions. ATP-DEC (green line) captures this trend, but the normal TIM-based method (red line) fails to adapt, consistently underestimating emissions. The dotted yellow line shows the theoretically TIM-based method scenario, where it incorporates the actual post-flight distance. Even in this theoretical scenario, ATP-DEC has a higher estimation. This is due to the extensive scope of ATP-DEC, incorporating factors such as in-flight services, aircraft and airport factors. The disparity between the two TIM-based scenarios is evident, with the method lacking actual distance data underreporting per passenger emissions by 23150 tonnes $CO_2$e across the 473 analysed flights in 2023. In contrast, ATP-DEC, which incorporates a 30-day rolling average HAF, exhibits a slight overestimation by just 51 tonnes $CO_2$e.

## Accounting for changes in-flight route using HAF

To demonstrate the impact of HAFs on capturing real-world flight variations, a case study was conducted on British Airways (BA) flights between London Heathrow (LHR) and Shanghai Pudong (PVG) for the years 2019 and 2023. These flights provide a clear example of how external events, such as the closure of Russian airspace, influence flight routes, distances, and carbon emissions.

Case Study Setup:

- 2019 Flights: Represent pre-pandemic operations without major disruptions. However, even in this period, flights show minor

variations in flight distances and mean latitudes due to operational constraints and minor route inefficiencies.

- 2023 Flights: BA resumed LHR-PVG flights in April 2023 after a pandemic-related suspension. Upon resumption, flights were rerouted to avoid Russian airspace due to the ongoing conflict, resulting in considerably longer flight paths and altered mean latitudes.

The analysis compares carbon footprint estimates using ATP-DEC with and without HAF, alongside post-flight records. Figures 3 and 4 illustrate the changes in-flight distance and mean latitude, while Fig. 5 demonstrates the resulting impact on carbon footprints for economy-class passengers. Figure 3 demonstrates that actual flight distances in 2023 were substantially longer compared to 2019 due to the rerouting required to avoid Russian airspace. Despite the absence of such disruptions in 2019, those flights also exhibit higher distances than the great circle distance (shown as the red line) (Equation S17). Static methods that rely solely on great circle distances fail to capture these variations. By incorporating historical flight path data, the HAF successfully adjusts for these factors, enabling the ATP-DEC to produce more realistic and precise distance estimates (Equation S8).

Changes in mean latitude, shown in Fig. 4, further highlight the impact of rerouted flights. The 2023 flights, which required a southward deviation to bypass Russian airspace, exhibited a reduced mean latitude compared to 2019. These shifts are particularly relevant for calculating non-Kyoto impacts, as atmospheric conditions vary by latitude, influencing the formation of $NO_x$, contrails, and water vapour (Equation S1–S3). Even for 2019 flights, where rerouting was not a factor, variations in mean latitude were

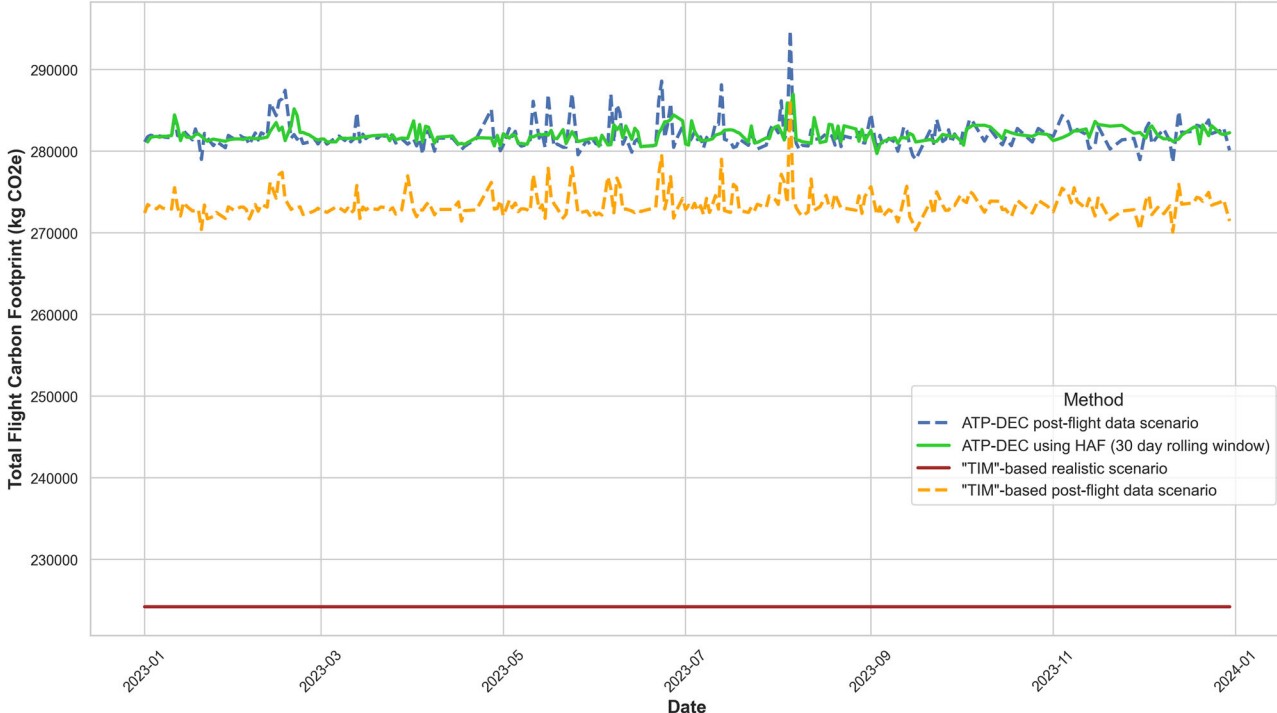

**Fig. 2 | Emission footprint method comparison for LHR-ICN flights in 2023 without non-Kyoto impacts[30].** Line plot showing total flight emissions footprint (kg $CO_2e$) for four methodological scenarios: ATP-DEC using historical adjustment factors (HAF, 30-day rolling window; green solid line), ATP-DEC using actual post-flight distance (blue dashed line), TIM-based method using constant distance (red solid line), and TIM-based method using actual post-flight distance (orange dashed line). The ATP-DEC HAF approach closely matches the emissions trend from post-flight data, capturing operational variations such as rerouting due to Russian airspace closure. The TIM-based constant-distance method consistently underestimates emissions, with a total underreporting of 23150 tonnes $CO_2e$ across 473 flights. In contrast, ATP-DEC with HAF slightly overestimates by just 51 tonnes $CO_2e$.

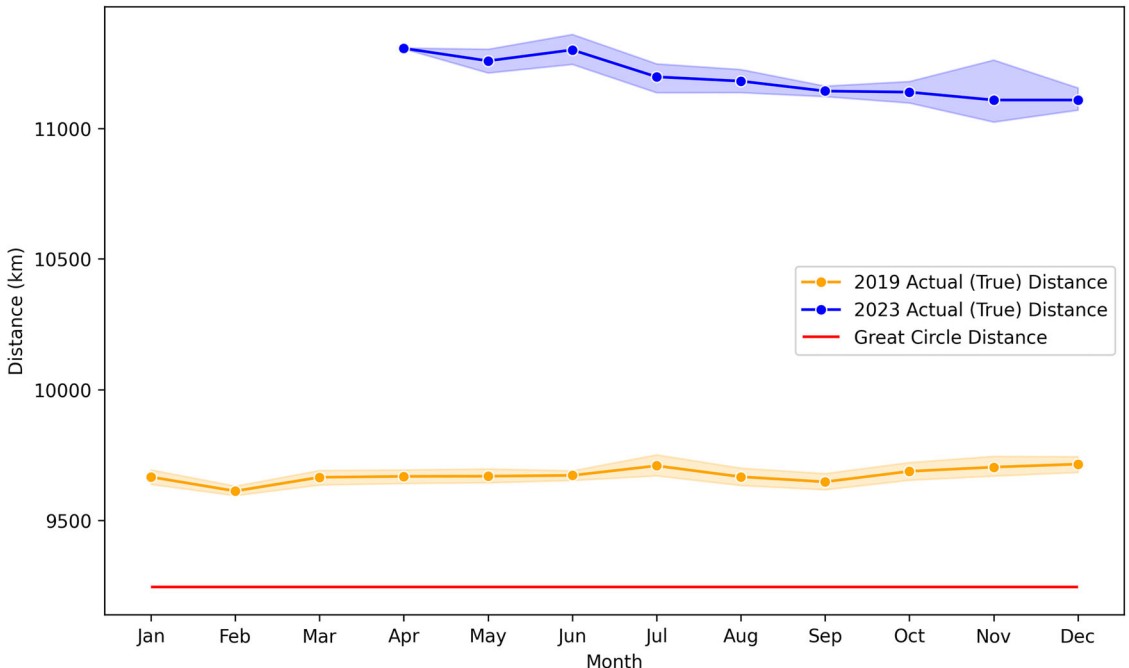

**Fig. 3 | Actual and great circle distances for British Airways flights between LHR and PVG in 2019 and 2023[30].** Monthly mean flight distances (km) for 2019 (orange circles and line with shaded range) and 2023 (blue circles and line with shaded range) compared with the great circle distance (red solid line). Shaded areas represent the monthly range of distances. The 2023 flights, resumed in April after the pandemic, were rerouted to avoid Russian airspace, resulting in longer distances than in 2019. Even in 2019, actual distances exceeded the great circle distance due to operational constraints.

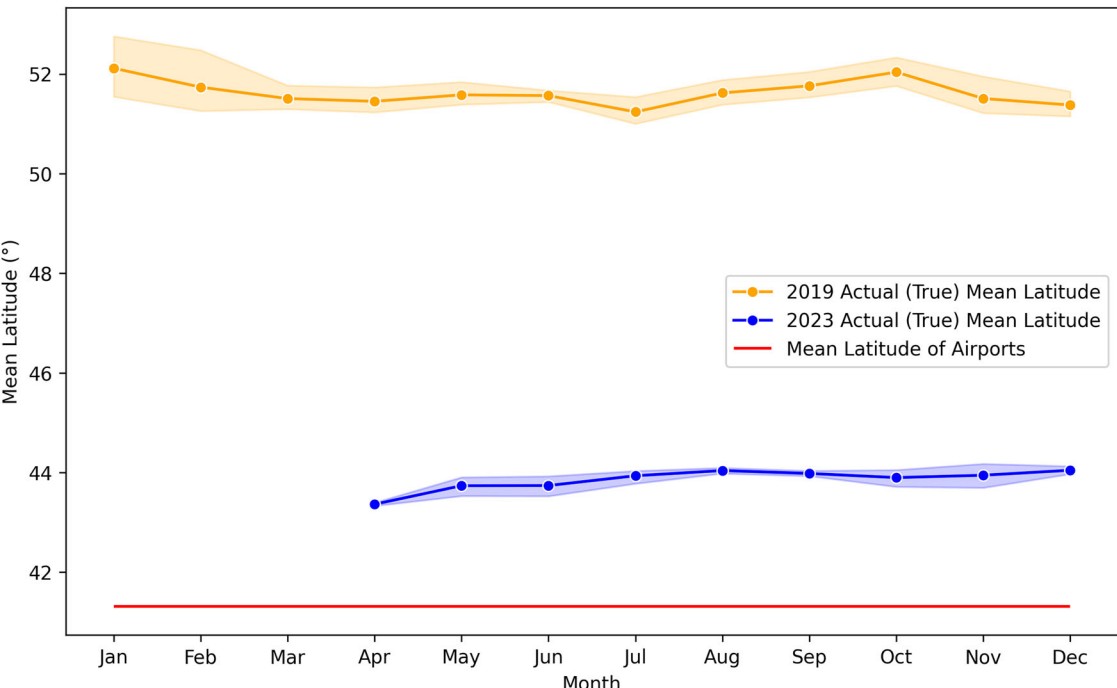

**Fig. 4 | Mean latitude for British Airways flights between LHR and PVG in 2019 and 2023[30].** Monthly mean flight latitudes (°) for 2019 (orange circles and line with shaded range) and 2023 (blue circles and line with shaded range) compared with the mean latitude of departure and arrival airports (red solid line). Shaded areas represent the monthly range of latitudes. In 2023, rerouting to avoid Russian airspace resulted in a lower mean latitude compared to 2019, reflecting a southward deviation in flight paths. Even in 2019, actual latitudes varied month to month due to operational adjustments.

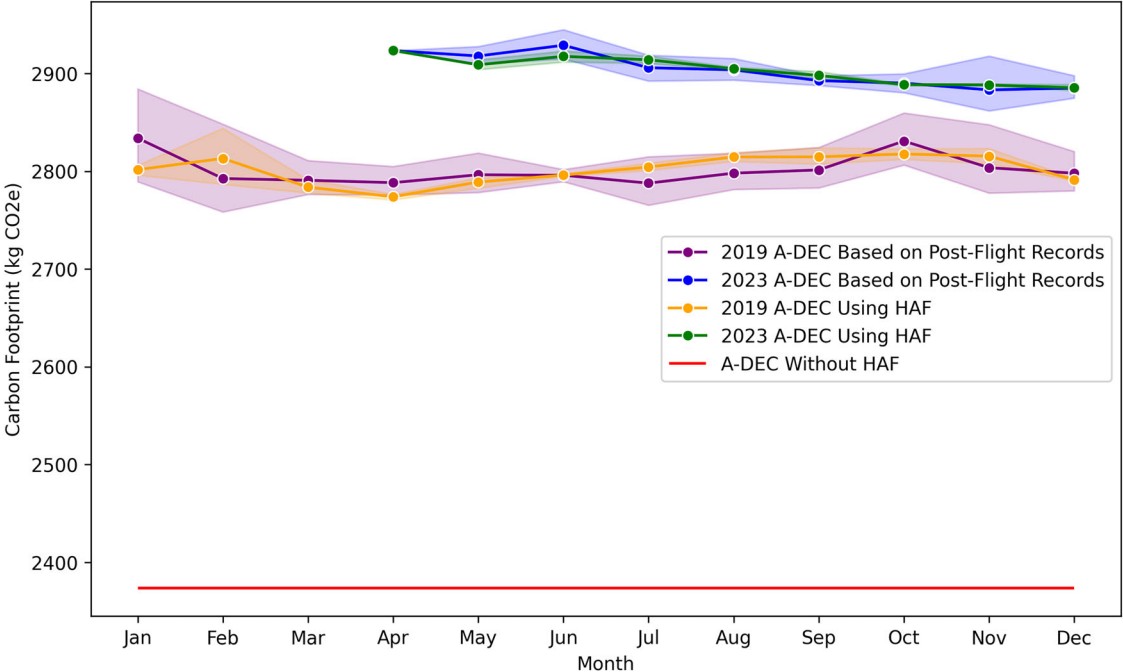

**Fig. 5 | Emission footprint estimates for economy-class passengers on British Airways flights between LHR and PVG in 2019 and 2023[30].** Monthly emissions footprint (kg $CO_2$e) estimated by ATP-DEC using historical adjustment factors (HAF; 2019: orange circles and line, 2023: green circles and line), ATP-DEC based on post-flight records (2019: purple circles and line, 2023: blue circles and line), and ATP-DEC without HAF (red solid line). Shaded areas represent the monthly range of values. In 2023, rerouting to avoid Russian airspace led to higher emissions compared to 2019. The HAF approach closely matched post-flight estimates in both years, whereas the static-distance method without HAF consistently underestimated emissions.

observed due to operational adjustments. The HAF dynamically captures these changes, ensuring that latitude-dependent emissions estimates reflect real-world conditions rather than static assumptions, such as using the mean latitude of airports (red line).

The results clearly show the ability of HAF to account for changes in-flight behaviour and improve the accuracy of emissions estimates. The impact of these changes on emissions is evident in Fig. 5. The longer distances and altered latitudes in 2023 resulted in a higher carbon footprint for

**Table 1 | Mean squared percentage error (MSPE) for pre-adjustment, 7-day HAF, and 30-day HAF methods across selected flight routes in 2019 and 2023[30]**

| Route | 2019 | | | 2023 | | |
|---|---|---|---|---|---|---|
| | Post 30 | Post 7 | Pre | Post 30 | Post 7 | Pre |
| BKK-CPH | 1.15 | 0.76 | 6.07 | 0.55 | 0.54 | 11.69 |
| CPH-BKK | 1.58 | 1.07 | 7.38 | 0.64 | 0.69 | 12.4 |
| FRA-ICN | 1.5 | 1.47 | 15.68 | 0.49 | 0.5 | 15.69 |
| ICN-FRA | 2.05 | 2.02 | 17.61 | 0.42 | 0.42 | 16.76 |
| ICN-LHR | 1.93 | 1.85 | 20.4 | 0.42 | 0.43 | 18.85 |
| LHR-ICN | 1.14 | 1.16 | 17.06 | 0.37 | 0.38 | 17.34 |
| LHR-PVG | 0.98 | 1 | 14.91 | 0.81 | 0.79 | 16.47 |
| PVG-LHR | 1.29 | 1.28 | 15.84 | 0.71 | 0.69 | 17.47 |
| SIN-ZRH | 1.41 | 1.21 | 2.04 | 1 | 0.9 | 1.52 |
| ZRH-SIN | 1 | 0.87 | 2.63 | 1.07 | 1.13 | 2.13 |

Bold in the column headers is for category grouping only and does not indicate statistical significance.

economy-class passengers, as seen in the post-flight estimates (blue line). Using HAF (green line) aligns the estimations with these post-flight values, demonstrating its ability to account for both large-scale disruptions and routine operational variations. In contrast, ATP-DEC without HAF (red line) consistently underestimates emissions by assuming static flight distances and latitudes. Even for 2019, the HAF enhances the accuracy of the ATP-DEC by capturing subtle variations in route efficiency, which static models overlook.

## Analysis of HAF rolling window length

The impact of the HAF rolling window length was assessed using 7-day and 30-day windows. Table 1 summarises the mean squared percentage error (MSPE) for each flight route across the dataset, comparing pre-adjustment results with post-adjustment performance.

The pre-adjustment method, which does not incorporate any historical data, consistently produces the highest MSPE values. This highlights the substantial estimation errors introduced when historical trends in route variability, delays, and operational adjustments are ignored.

The results indicate that the 7-day rolling window generally achieves the lowest MSPE values, providing superior short-term precision. This is particularly evident for flight routes with high variability, where a shorter window more effectively captures recent operational trends, such as sudden changes in delays or rerouting. For these routes, the 7-day HAF quickly adapts to fluctuations, reducing estimation errors.

By contrast, the 30-day rolling window shows slightly higher MSPE values for short-term variability but demonstrates competitive performance, especially for routes exhibiting more stable trends or pronounced seasonality. For the cleaner and less noisy 2023 flight data, the 30-day window even marginally outperformed the 7-day window over the full year. This suggests that, for long-term trends and less-volatile routes, a longer window can smooth transient anomalies and offer more precise results.

## Discussion
### Evaluation of ATP-DEC against existing tools

Table 2 presents a condensed comparative analysis of the ATP-DEC against TIM, IATA, ICAO, and MyClimate. The comparison is structured around the four core evaluative criteria: scope, consistency and accuracy, transparency, and effectiveness of communication (Table 2).

## Strengths and innovation of ATP-DEC

**Granularity of emissions breakdown.** One of the defining strengths of ATP-DEC is its ability to decompose emissions into distinct sources, such as TTW, WTT, carry-on and checked luggage, non-Kyoto impacts ($NO_x$, $H_2O$, CiC), and life cycle emissions from in-flight services, airports, and aircraft. This allows users to pinpoint emission sources and prioritise effective interventions. By offering a comprehensive view of aviation emissions, ATP-DEC supports both informed decision-making and targeted environmental action. ATP-DEC's modular framework allows users to include or exclude specific variables; the tool is adaptable to a variety of contexts.

Furthermore, ATP-DEC is modular, so it's flexible to evolving science and data. Each individual constituent of ATP-DEC can be reworked without affecting the others. This also encourages advancement. For example, the default value for fuel emission factors is Jet A1 fuel. But this can be easily altered when SAF or electric aircraft are used in the future.

**Inclusion of non-Kyoto impacts.** By incorporating non-Kyoto effects, the ATP-DEC provides a more comprehensive assessment of aviation's climate impact. This includes dynamic modelling of $NO_x$, $H_2O$, and CiC, which are often overlooked or oversimplified in existing calculators. The ability to accurately estimate these emissions aligns the ATP-DEC with evolving scientific understanding, filling a critical gap in current methodologies[15].

**Real-world adaptability.** Through the integration of HAFs, ATP-DEC captures real-world variations in-flight operations, such as rerouting due to airspace closures or delays. Static models lack this adaptability. The inclusion of rolling windows for HAF further enhances its ability to track both short-term fluctuations and long-term trends.

Validation of this approach was carried out using a dataset of operational flight records that included the actual flown distance and the actual mean latitude for each flight. For each day in the dataset, we compared the estimated GCD and estimated mean latitude (calculated purely from origin-destination coordinates) with the corresponding observed values. From these comparisons, an HAF was computed for both distance and mean latitude by taking the ratio of actual to estimated values of each flight. For example, a distance HAF of 1.05 indicates that actual flight paths are ~5% longer than the direct geographic estimate due to routing constraints, weather avoidance, or air traffic control. The adjustment factors were then applied to the estimates of subsequent flights to better approximate real operational conditions. To stabilise the correction and reduce the impact of outlier days, the HAFs were computed using rolling windows of either 7 or 30 days, ensuring they remained responsive to recent trends while retaining generalisability. The results indicated that the 7-day window was more accurate and will be adopted as the default for the model going forward.

**Improved passenger class weighting.** The ATP-DEC's approach to seat class weighting represents another improvement over existing tools. By utilising airline and aircraft-specific seating data, the ATP-DEC ensures fair and realistic attribution of emissions.

**Transparency and actionability.** The transparency of the ATP-DEC enhances its utility for stakeholders. By providing detailed outputs that clearly attribute emissions to specific sources, the tool facilitates accountability and supports actionable insights. The publication of the methodology allows stakeholders to fully understand their footprint.

## Technological integration and scalability

The ATP-DEC methodology is designed for seamless technological integration and scalability, ensuring its applicability across diverse operational contexts. The required input data is already widely available through existing industry sources. Furthermore, its modular design allows adaptation when certain data variables may be missing or incomplete. This

**Table 2 | Condensed carbon calculator review of four existing tools with a comparison to the ATP-DEC[30]**

| | Google travel impact model (TIM) 3.0–Accessed: 25 Jun 25 | IATA CO$_2$ connect – Accessed: 28 Feb 24 | ICAO carbon emissions calculator – Accessed: 11 Jun 25 | MyClimate 1.0.1 Accessed: 11 Jun 25 | ATP-DEC – Accessed: 25 Jun 25 |
|---|---|---|---|---|---|
| Scope | **Low-mid**: includes all Kyoto GHG and WTT and TTW emissions, includes contrail "buckets" but no quantitative assessment, adds static airport emissions | **Low**: includes only CO$_2$ emissions | **Low**: only includes CO$_2$ emissions, only offers two possible passenger classes | **Mid-high**: includes non-CO$_2$ emissions, freight allocation, and most upstream emissions (no in-flight services) | **High**: includes non-Kyoto effects, upstream emissions, cargo and passenger luggage allocation |
| Consistency & accuracy | **Mid**: uses reputable data sources from outside organisations but no historical adjustment, makes static assumptions for airport emissions and distance adjustment | **Mid**: uses accurate industry-sourced data but no historical adjustment | **Low**: claims to use industry data but calculator output is far lower than competitors and no historical adjustment | **Low-mid**: uses lesser-known data sources and no historical adjustment | **High**: uses high-level estimation models for distance and fuel consumption and historical adjustment to refine estimates |
| Transparency | **Mid-high**: well-documented methodology and easy to access, contrail bucket method classes impact as "high", "medium", "low", which gives little transparent indication with the vast ranges between them | **Low-mid**: limited documentation with no technical explanation | **Low**: easily accessible documentation with basic explanation | **Mid-high**: detailed methodology, easily accessible | **High**: detailed methodology accessible through this article and supplementary methods, also outputs a full breakdown of emissions sources |
| Effectiveness of communication | **Mid**: Suggestions to select alternate flights based on emission comparison, no options to offset or suggestions to facilitate, no breakdown of emissions | **Low-mid**: user-friendly and clear display but full service is paid, no options to offset or suggestions to facilitate, no breakdown of emissions | **Low**: confusing user interface and offers little to no understanding of the meaning of the footprint, no options to offset or suggestions to facilitate, no breakdown of emissions | **Mid**: clear output with options to offset footprint using a range of schemes, no breakdown of emissions | **High**: full breakdown of emissions sources and linked to transparent, robust, blockchain-based carbon offsetting mechanism |
| Overall assessment | **Mid**: aligns with ISO 14083 but fails to include all non-CO$_2$ impacts and some upstream emissions | **Low-mid**: excludes non-CO$_2$ GHG emissions and upstream emissions | **Low**: flawed method and outdated passenger load factors and class weightings | **Mid-high**: references up to date and scientific approaches but non-Kyoto effects are approximated, and the method makes generalised assumptions | **High**: scientific, data-driven approach with dynamic modelling of all emission sources, providing full breakdown for simple understanding |

Bold in the column headers is for category grouping only and does not indicate statistical significance. Bold in qualitative ratings ("Low", "Low-mid", "Mid", "Mid-high", "High") is for emphasis only and does not indicate statistical significance.

adaptability ensures that the tool remains functional, making it particularly useful for regions or sectors with variable data availability.

The deployment of the ATP-DEC is straightforward, with its architecture allowing easy integration into existing software systems, such as airline booking platforms or carbon offsetting programs. Its design supports scalability, enabling implementation across multiple flights or operational networks without much computational or logistical overhead.

A key feature of the ATP-DEC is its symbiotic relationship with a blockchain-based carbon offsetting mechanism. This integration facilitates transparent and verifiable environmental action, allowing stakeholders to directly link emissions calculations with projects with a documented positive environmental impact. By leveraging blockchain technology, the methodology ensures traceability and accountability, enhancing trust among stakeholders. The combination of accurate emissions assessments and seamless offsetting ecosystem positions the ATP-DEC as an essential tool for aligning aviation practices with global climate goals.

## Model limitations and challenges

**HAF forecast.** The HAF provides dynamic emissions adjustments based on historical trends, making it uniquely suited for real-time applications. However, for flights booked in advance, the HAF is calculated on the day of booking using the most recent data available. This ensures that the emissions estimate reflects the latest operational conditions, such as route deviations or airspace restrictions at the time of booking. The HAF is not yet suitable for forecasting future flight conditions.

**Engine efficiency.** The methodology currently considers aircraft type and age, but does not account for the specific engines installed on each aircraft. Since engines are often replaced or upgraded independently of the airframe, this omission may affect the accuracy of fuel burn and emissions estimates, particularly for older aircraft retrofitted with more efficient engines.

**Exclusion of soot and sulphate aerosols.** While the ATP-DEC's non-Kyoto impacts modelling incorporates contrail-induced cloudiness (CiC), it does not explicitly account for the impacts of soot and sulphate aerosols. Although sulphate aerosols are thought to exert a cooling effect, their exclusion, along with soot's warming contributions, reflects a limitation in the Dahlmann et al. method used in the ATP-DEC[22]. While soot is a contributor to global warming, sulphate aerosols have a cooling effect. With comparable magnitude, sulphate aerosols help to offset soot's warming effect. Their modelling uncertainty has been recognised in the literature[24].

**Non-Kyoto impacts uncertainty.** The scientific understanding and modelling of non-Kyoto impacts is still rapidly evolving. While the Dahlmann et al. method was developed based on the emission profile of A330-200, the most commonly sold medium and long-range aircraft[22], other aircraft could produce different results. Dahlmann et al.[20] calculate that their method produces a mean squared error of 0.19 compared to advanced contrail modelling of specific flights using detailed flight data, whereas using a constant multiplier to assess non-Kyoto emissions produces a mean squared error of 1.18 for these flights. Dahlmann et al.'s[22] method remains the most credible for estimating non-Kyoto impacts in terms of $CO_2e$ but is constrained by the available scientific data.

**Scalability of contrail modelling.** Advanced contrail modelling solutions exist that incorporate vast amounts of real-time atmospheric data, offering higher precision. However, these methods are computationally intensive and unfeasible for flight-by-flight calculations at scale. As a result, the ATP-DEC adopts a practical approach that balances accuracy with computational efficiency, recognising that more granular contrail modelling would require considerable technological advancements.

**Sensitivity of variables.** A sensitivity analysis was outside the scope of work for this study, but it's acknowledged as an important aspect of future work. Evaluating how variations in key input parameters affect model output can help identify dominant variables and better characterise model uncertainty. Various methods can be used, including both deterministic and probabilistic single or multi-variable approaches. Data uncertainty can be managed by probabilistic or stochastic approaches, such as Monte Carlo simulation and some deterministic ways, such as scenario analysis and sensitivity analysis[25].

## Conclusions

The ATP-DEC methodology represents a substantial advancement in aviation carbon footprint calculation, offering superior accuracy, scope, and real-world applicability. It sets a new benchmark for addressing the complex challenges of emissions accountability while fostering transparency and actionable insights.

### Superior methodology

By incorporating advanced modelling techniques such as HAFs, which utilise historical flight data to achieve exceptional accuracy. Unlike existing tools, it includes an expansive scope that covers the full life cycle of emissions. This footprint breakdown allows stakeholders to understand their environmental impact in a more meaningful way, promoting effective communication and targeted interventions. Furthermore, the transparency of the methodology, demonstrated through its detailed publication, ensures trust and credibility.

### Benchmark for future standards

Current standards, such as ISO 14083, remain constrained by limited scope and accuracy. However, the data required to dramatically improve these estimations already exists, as demonstrated by ATP-DEC's ability to match real-world trends. This methodology not only highlights the inadequacies of existing frameworks but also sets a precedent for the adoption of more comprehensive and scientifically informed standards.

### Nudge to environmental action

ATP-DEC is designed to drive environmental action. Its ease of deployment ensures that it can be readily integrated into existing operational systems, leveraging widely available data. This accessibility makes it a valuable component of broader sustainability strategies. While the ATP-DEC itself is not the sole solution to aviation emissions, it functions as a critical part of the ecosystem, working symbiotically with initiatives such as blockchain-based carbon offsetting mechanisms[21]. This integration facilitates traceable and verifiable environmental actions, empowering stakeholders to engage in meaningful climate mitigation efforts.

In conclusion, the ATP-DEC methodology represents a transformative approach to aviation emissions accounting. By combining superior accuracy, extensive scope, and actionable insights, it surpasses existing tools while setting a foundation for future standards. Its potential to inspire confidence, inform policy, and drive sustainable practices underscores its critical role in addressing the aviation industry's contribution to global climate goals.

## Methods
### Approach overview

ATP-DEC follows an LCA approach in line with ISO 14083, quantifying GHG emissions per passenger for a given flight. The cradle-to-grave system scope in the aviation sector's context includes the production and transportation of fuel WTT, fuel burning during flying, take-off and landing TTW, in-flight services, and airport and aircraft life cycles (Table S1). In addition, ATP-DEC incorporates non-Kyoto impacts specific to aviation. Emission allocation incorporates luggage emissions, aircraft-specific seating configurations, passenger load factors, and cargo load factors. The goal of the tool is to ensure no source of climate impact is overlooked while dynamic

models, specific inputs, and life cycle data are combined to deliver high accuracy. HAFs refine pre-flight estimates by integrating real-world variations.

## Interpretation: ATP-DEC's step-by-step calculations

This section summarises the step-by-step calculations to interpret the results, specifically the hotspot analysis of a flight from Singapore to Zurich (Fig. 1). Equation 1 summarises the methodology, with variables described in Table S5. Further justification of each individual variable and how it's calculated can be found in Section 1.2 of the Supplementary Information (SI).

$$\text{CO}_2\text{e per pax} = \mathcal{G}\left(\frac{f(\text{D}) \times (\text{WTT} + \text{TTW}) \times \text{DT} \times \text{CF} \times \text{LF} \times \text{CW}}{\text{PLF}} + L + (V \times \text{IFS}) + \text{APF} + \text{AF} + (\mathcal{Z}_1 \times \text{N}(\mathcal{Z}_2))\right)$$

(1)

For steps 1–6 the scope aligns with ISO 14083 and provides a robust, standardised framework for emission calculations. The extended methodology, 7–9, supplements the method, elevating comprehensiveness. The additional and enhanced factors within the extended methodology go beyond the scope of ISO 14083 and are optional. The ATP-DEC methodology outlines a full, holistic, modular formula that acknowledges the importance of a thorough environmental assessment while allowing flexibility for the stakeholder to encourage environmental action.

### Calculate distance

Calculate the initial distance using the GCD and HAF (Equation S8).

For Fig. 1 (Singapore to Zurich), the GCD is 10,309 km, calculated using the geographical coordinates of the airports. An HAF can be used to adjust this to a more precise flight distance (Equation S13). For Fig. 1, historical flight data from the arbitrarily chosen final week in December 2019 were used to calculate a distance HAF of 1.0315 (Equation S12 and S19). This means that the average distance of a flight in this month was 1.0315 times higher than the GCD, and we use this to predict the distance of the flight example in Fig. 1 as 10,634 km. The HAF can be updated with a rolling window of 7 or 30 days. For Fig. 1 a rolling window of 7 is used because it proved to be the most accurate in capturing trends (Table 1).

### Extract fuel burn

Use a polynomial regression model alongside the fuel burn data to equate the distance flown with the aircraft-specific fuel consumption. Include both the landing, taxi and take-off (LTO) and cruise, climb, descent (CCD) fuel consumption. Multiply by age deterioration factor to account for aircraft efficiency[26]. Where exact fuel burn is known, this value should be used instead.

For the results in this study, Eurocontrol fuel data is used[27]. Fuel burn for 10,634 km for a Boeing 777–300 (B773) is 89,540 kg (Fig. S1). For a wide-bodied B773 with an age of 1 year, the deterioration factor is 1.01[26], so the fuel consumption is given as 90,435 kg (Fig. 1).

### Compute total aircraft emissions

Multiply the fuel consumption by the WTT and TTW emissions factors and sum them together to calculate the total fuel emissions for the flight. This includes all passengers, passenger luggage, and cargo emissions.

The WTT and TTW emissions for 90,435 kg of Jet A1 fuel are 43,409 kg $\text{CO}_2$e and 28,5776 kg $\text{CO}_2$e, respectively. This is calculated using emission factors from ISO 14083[28] (Equation S20).

### Adjust for cargo and passenger luggage allocation

For emission accounting, the total mass of the aircraft can be assumed to be the sum of passengers, passenger luggage, and cargo (freight and mail). After

subsequently calculating the cargo and luggage factors, the total emissions are adjusted to account solely for passengers.

Cargo mass is assumed to be zero for the results in Fig. 1, to allow fair comparison. Carry-on and checked luggage are accounted for individually to enable a full breakdown of emissions. The average mass of passengers and luggage by class is sourced from EASA[29] (Table S3). The HAF-enhanced passenger load factor (PLF) of 0.85 is used when estimating the mass of each constituent (Equation S5, S22–S26). Following this, the luggage and cargo factors can be calculated and the passenger emissions adjusted (Equation S27–S29). The total passenger WTT emissions and TTW emissions (for a fully occupied example flight (Fig. 1)) are 33,147 kg and 218,219 kg $\text{CO}_2$e, respectively (Equation S30, S31).

### Compute class weightings

Class weighting is calculated by dividing the area of a single seat in each class by the total seat area of the aircraft (Equation S10). Seat data for the example flight (Fig. 1) is acquired from SeatGuru[23] (Table S6) (Equation S32), although this can normally be sourced through airlines for specific flights. The total $\text{CO}_2$e is multiplied by the class weighting (Table S7) to compute the mass of $\text{CO}_2$e for each passenger in each class (for 100% PLF) (Equation S11) (Table S8) (Fig. 1).

### Adjust for empty seats

The WTT and TTW emissions per passenger are finally determined by dividing the total passenger allocated WTT and TTW emissions by the PLF of the flight to account for empty seats (Table S9).

### Calculate passenger luggage emissions

Total passenger luggage allocated $\text{CO}_2$e emissions are given by removing the passenger and cargo emissions. This separation allows for individual calculation of luggage carbon footprint. This allows passengers to select the luggage that they wish to take and can encourage more sustainable choices. For the results in Fig. 1, Table S10 shows luggage emissions. It's important to note that the carry-on and checked luggage emissions in this table represent the full comprehensive output, including non-Kyoto impacts as well as WTT and TTW (Fig. 1).

### Add in-flight service, airport and aircraft factors

Add the in-flight service constant dependent on the passenger class and add the airport and aircraft constants dependent on the distance of the flight (Table S4).

### Non-Kyoto impacts

The output of the non-Kyoto formulae is multiplied by the $\text{CO}_2$ emissions per passenger, which must be calculated individually, using a very similar method to steps 3.3.1 to 3.3.6, replacing the WTT and TTW emissions factors with the $\text{CO}_2$ emissions factor of 3.15 kg $\text{CO}_2$ per kg of fuel[22]. This step is taken to prevent double-counting of emissions, since Kyoto gases must be removed for the non-Kyoto impact of $\text{NO}_x$, $\text{H}_2\text{O}$, CiC.

For the Singapore to Zurich example, the HAF-refined distance is 10634 km. Converted to thousand kilometres, this is 10.634. The mean latitude between the airports is 24.4°, but the HAF-refined mean latitude is 28.4° based on historical operational latitude data.

These values are substituted into the non-Kyoto equations (Equation S1–S3) (Table S2) to compute the non-Kyoto multipliers of 1.46, 1.42, and 0.20 for $\text{NO}_x$, CiC, and $\text{H}_2\text{O}$, respectively (Equation S33–S35). Multiplying by the per passenger $\text{CO}_2$ gives the non-Kyoto impact for each class (Equation S4) (Table S11).

### Inclusion & ethics

This research did not involve human participants, animal subjects, or personally identifiable data, and therefore did not require institutional ethics approval. The development of the ATP-DEC model was carried out with a commitment to transparency, reproducibility, and environmental responsibility. We have aimed to include diverse perspectives in the development of

the model by engaging with a range of literature and datasets from multiple geographic regions and sources. All collaborators contributed meaningfully to the research and authorship, and efforts were made to ensure an inclusive and respectful research environment throughout the study.

## Reporting summary
Further information on research design is available in the Nature Portfolio Reporting Summary linked to this article.

## Data availability
The development of the ATP-DEC model drew upon both publicly available data sources and licensed third-party data. Public data used in the methodology includes emissions factors, aircraft specifications, and life cycle inventory data from sources such as the EEA, airlines, and relevant academic literature. For the simulation and analysis of ATP-DEC, the study used historical flight operations data purchased from FlightRadar24. This data is not publicly available. Key derived results are presented within the manuscript, and the data required to interpret, replicate and build upon the methods or findings can be found at https://figshare.com/articles/dataset/Figure_Data_xlsx/29852306?file=57008798[30]. The core code required to replicate results can be found at https://doi.org/10.5281/zenodo.17196593[31].

## Code availability
The ATP-DEC calculator core, example processing stubs, and figure-generation scripts are openly available at Zenodo (https://doi.org/10.5281/zenodo.17196593) under the AGPL-3.0-only license. Proprietary raw flight datasets cannot be shared; some schema-identical synthetic data and all the output data are included to enable reproducibility of figures and results[31].

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

## Acknowledgements
The authors gratefully acknowledge the support of Innovate UK (grant number 10076949). This work was part-funded by the UK Research and Innovation Fund through Innovate UK and part-funded by Therme Group RHTG AG through an Innovate UK Knowledge Transfer Partnership (grant number 10076949). For the purpose of open access, the authors have

applied a Creative Commons Attribution (CC BY) licence to any Author Accepted Manuscript version arising from this submission.

## Author contributions

F. McFall: methodology, software, validation, investigation, data curation, writing–original draft, visualisation, formal analysis. D. Guan: methodology, resources, writing–review & editing. J. Chenoweth, X. Font, J. Sadhukhan: methodology, validation, investigation, writing–original draft, supervision. I. Corduneanu: conceptualisation, investigation, methodology, resources, writing–review & editing, project administration, funding acquisition. E. Goean: conceptualisation, methodology, resources, writing–review & editing, supervision, project administration, funding acquisition.

## Competing interests

I. Corduneanu and E Goean are employees of Therme Group. E. Goean holds a management position as a board member at Therme Group, serving as VP of New Business Partnerships. The University of Surrey has received funding from Therme Group RHTG AG, which aims to commercialise the carbon calculator. All other authors declare no competing interests.
