## [Transparent Peer Review file · Communications Earth & Environment]

Aviation Passenger Carbon Footprint Calculator with Comprehensive Emissions, Life Cycle Coverage, and Historical Adjustment

Corresponding Author: Mr Finn McFall

Version 0:

Decision Letter:

Dear Mr McFall,

Your manuscript titled "Air Travel Passenger Dynamic Emissions Calculator (**ATP-DEC**): A carbon footprint calculator for aviation with full LCA & historical adjustment factors" has now been seen by 3 reviewers, and we include their comments at the end of this message. They find your work of interest, but some important points are raised. We are interested in the possibility of publishing your study in Communications Earth & Environment, but would like to consider your responses to these concerns and assess a revised manuscript before we make a final decision on publication.

We invite you to revise your submission and resubmit to Communications Earth and Environment a presentation of a calculation tool for a broad range of aviation emissions that relies on compelling comparison criteria such as international ISO standards. This work should provide clear methodological details, in the main manuscript, necessary to reproduce the findings. It should also assess and report transparently limitations and uncertainties.

We therefore invite you to revise and resubmit your manuscript, along with a point-by-point response that takes into account the points raised. Please highlight all changes in the manuscript text file.

Please submit your point-by-point responses as a separate file, distinct from your cover letter where you can add responses to the Editors' comments that you do not want to be made available to the reviewers. Word files are preferred. We recommend that any figures, tables or graphs that are included in the response to reviewers are also included in the main article or Supplementary Information.

Please use the following link to submit your revised manuscript, point-by-point response to the referees' comments (which should be in a separate document to any cover letter), a tracked-changes version of the manuscript (as a PDF file) and the completed checklist:

Link Redacted

We hope to receive your revised paper within six weeks; please let us know if you aren't able to submit it within this time so that we can discuss how best to proceed. If we don't hear from you, and the revision process takes significantly longer, we may close your file. In this event, we will still be happy to reconsider your paper at a later date, as long as nothing similar has been accepted for publication at Communications Earth & Environment or published elsewhere in the meantime.

Please do not hesitate to contact us if you have any questions or would like to discuss these revisions further. We look

forward to seeing the revised manuscript and thank you for the opportunity to review your work.

Best regards,

Nandita Basu, PhD
Consulting Editor, Communications Earth & Environment
Associate Editor, Communications Sustainability
Nature Portfolio

EDITORIAL POLICIES AND FORMATTING

Editorial Policy: [Policy requirements](https://www.nature.com/documents/nr-editorial-policy-checklist.pdf) (Download the link to your computer as a PDF.)

- Behavioural and social science
- Ecological, evolutionary & environmental sciences
- Life sciences

<https://www.nature.com/documents/nr-reporting-summary.zip>

Furthermore, please align your manuscript with our format requirements, which are summarized on the following checklist: [Communications Earth & Environment formatting checklist](https://www.nature.com/documents/commsj-phys-style-formatting-checklist-article.pdf)

and also in our style and formatting guide [Communications Earth & Environment formatting guide](https://www.nature.com/documents/commsj-phys-style-formatting-guide-accept.pdf).

*** DATA: Communications Earth & Environment endorses the principles of the Enabling FAIR data project (<http://www.copdess.org/enabling-fair-data-project/>). We ask authors to make the data that support their conclusions available in permanent, publically accessible data repositories. (Please contact the editor if you are unable to make your data available).

All Communications Earth & Environment manuscripts must include a section titled "Data Availability" at the end of the Methods section or main text (if no Methods). More information on this policy, is available at <http://www.nature.com/authors/policies/data/data-availability-statements-data-citations.pdf>.

If a community resource is unavailable, data can be submitted to generalist repositories such as [figshare](https://figshare.com/) or [Dryad Digital Repository](http://datadryad.org/). Please provide a unique identifier for the data (for example a DOI or a permanent URL) in the data availability statement, if possible. If the repository does not provide identifiers, we encourage authors to supply the search terms that will return the data. For data that have been obtained from publically available sources, please provide a URL and the specific data product name in the data availability statement. Data with a DOI should be further cited in the methods reference section.

REVIEWER COMMENTS:

Reviewer #1 (Remarks to the Author):

I believe that the study will undoubtedly contribute to academic and sectoral analyses of the aviation sector and will yield more accurate results

Reviewer #2 (Remarks to the Author):

The article presents an innovative aviation emissions calculation tool, ATP-DEC. Compared to existing carbon emission calculators, this tool has a broader scope of emissions calculation; it also introduces a historical adjustment factor (HAF) for dynamic adjustments based on historical flight data; adopts a comprehensive Life Cycle Assessment (LCA) method, considering the entire life cycle emissions of aircraft; and takes into account passenger cabin classes, luggage, and cargo, allowing for more detailed emission allocations for different passenger groups. Based on this research, I have the following questions that I hope the authors will address :

1. The article briefly discusses several existing tools (such as TIM, IATA, ICAO, MyClimate). It might be helpful to focus more on ATP-DEC and compare it within the context of existing research to strengthen the argument for ATP-DEC's innovation in this field.
2. Some assumptions made in the model (such as excluding certain non-Kyoto impact factors like soot and sulfate aerosols, and the selection of the historical adjustment factor (HAF)) lack sufficient justification. A more detailed discussion is needed on why these emission factors or effects were excluded, especially the potential role of soot and sulfate aerosols, which could impact the results.
3. Although the article mentions that ATP-DEC was validated against 30,000 historical flights, there is no discussion of the model's uncertainty. A more formal uncertainty analysis should be included, particularly given the complexity of dynamic modeling. A sensitivity analysis is recommended for future work.
4. The article mentions that ATP-DEC can be applied globally, but lacks an in-depth discussion on its applicability in different regions. The aviation operation modes, aircraft types, flight frequencies, and climate conditions in different regions could all affect the emission calculation results. It is suggested to add an analysis of the model's applicability in different regions and highlight potential limitations.
5. While the article compares ATP-DEC with existing tools, there is no direct comparison with international standards such as ISO 14083 or IPCC guidelines. It would be beneficial to include a comparison with these international standards in terms of methodology, emission factors, and data usage, demonstrating how ATP-DEC aligns with or exceeds these industry standards.
6. The article primarily focuses on the analysis of existing flight data but lacks a predictive analysis of future aviation emission trends. For example, as the aviation industry transitions towards more environmentally friendly technologies (such as electric aircraft and sustainable aviation fuels), how will ATP-DEC adapt to these changes?
7. The article could also benefit from the inclusion of additional references to strengthen the background and contextualization of the research.

Reviewer #3 (Remarks to the Author):

The manuscript presents an innovative approach to improving carbon footprint calculators for passenger aviation. It incorporates non-Kyoto effects, upstream emissions, and lifecycle emissions from aircraft and airports. The manuscript introduces a comprehensive methodology that addresses key gaps in the current literature, particularly in terms of scope, accuracy, and transparency, making an important contribution to the field. Especially the integration of real flight data is a strong point and offers significant potential.

General Comments:

Overall, the manuscript is well-written, clearly structured, and easy to follow. The topic is highly relevant and offers valuable insights into improving carbon footprint assessments for aviation.

Detailed Comments:

- The abbreviations "MSPE" and "ATP-DEC" are used in the abstract without being introduced. It would be helpful to define them upon first use for better clarity.
- The terms "life cycle assessment" and "carbon footprint" are used somewhat interchangeably. However, LCA is a standardized, comprehensive method that evaluates various environmental impacts across multiple impact categories, while a carbon footprint specifically focuses on greenhouse gas emissions throughout a product's life cycle.
- The term "non-Kyoto emissions" could be confusing, as it implies a different context. I recommend the authors clarify this terminology or use the more widely accepted "non-CO₂ emissions" instead.
- The literature review is very short, and the gaps in existing research are not sufficiently highlighted. It would strengthen the paper if the authors could provide a more thorough review of relevant literature and more clearly articulate the gaps and their implications for the field. Current relevant papers in this field include:
 - Rupcic et al. (2023): Environmental impacts in the civil aviation sector: Current state and guidance. Doi:10.1016/j.trd.2023.103717
 - Cox et al. (2018): Life cycle assessment of air transportation and the Swiss commercial air transport fleet. Doi: 10.1016/j.trd.2017.10.017
 - Rahn et al. (2025): Quantifying climate impacts of flight operations: A discrete-event life cycle assessment approach. Doi:

10.1016/j.trd.2025.104646

- Keiser et al. (2023): Life cycle assessment in aviation: A systematic literature review of applications, methodological approaches and challenges. Doi: 10.1016/j.jairtraman.2023.102418

- The statement "The non-Kyoto effects of aviation on climate are substantially greater than the CO₂ effects" in the beginning of chapter 3 is very vague. I recommend providing quantitative data or more specific examples.
- The manuscript emphasizes the necessity of a full cradle-to-grave model to assess the environmental footprint of flights. However, the rationale for this approach is not clear. The authors should clarify what "cradle" and "grave" refer to in the context of aviation and why this model is vital for the assessment.
- The challenges in integrating non-CO₂ effects into assessments are only briefly mentioned. A more thorough literature review on why these effects are difficult to quantify (e.g., due to altitude, location, or other factors) would significantly improve the study's context and depth.
- The content and purpose of figure 1 is not completely clear to me. What is the idea behind this figure? The assumption of an aircraft with only first-class seats is unrealistic and I don't see the added-value in this graphic.
- The authors mention the often-overlooked emissions from luggage transportation. However, isn't this typically included within passenger-related emissions (PAX)? Clarification on this point would strengthen the argument.
- The tools mentioned are difficult for comparison as they do not include non-CO₂ effects. The authors should discuss this limitation more explicitly and explain how their approach provides a more comprehensive assessment.
- The structure of the use cases is unclear. The inclusion of different flights and time frames (e.g., pre-pandemic and pre-Russian occupation periods) lacks organization and makes the methodology hard to follow. A dedicated chapter explaining the methodology and choices made for these use cases/comparisons would greatly enhance clarity.
- The purpose and relevance of the rolling window length (chapter 4.4) are not clearly explained.
- What does DOI in table 2 mean?
- The methodology behind the tool is not sufficiently explained. How does it work and how can a user work apply it? A detailed description of how the tool operates and its underlying assumptions is crucial for understanding and evaluating its validity.
- The manuscript does not allow for reproducibility of the results, which is a fundamental aspect of scientific research. Clear documentation of methods and data sources is needed to address this issue. Additionally, the manuscript does not specify which database version or datasets were used. Including these details is essential for transparency and verification of the results.
- While the manuscript claims to consider "full life flight operations," it does not follow the standardized LCA methodology (based on ISO 14040/44). Key elements such as the definition of goals, scope, and inventory creation are unclear, which undermines the transparency of the study.
- The inclusion of non-CO₂ effects is presented as a unique selling point. However, the equations used appear to be entirely adapted from Dahlmann et al. (2023), where specific assumptions for their validity are explicitly stated. It is unclear whether these assumptions are actually met in the context of this manuscript. The authors should carefully evaluate whether the assumptions underlying these equations are satisfied and clearly justify their applicability to this study
- The supplementary material is very detailed, organized, and useful, but it is difficult to fully understand the paper without any explanation of the aspects presented in it. It would be helpful to clarify how and why these elements are relevant to the study within the main text, as this would improve the overall readability and comprehension of the paper.

While the manuscript presents an interesting approach and includes valuable insights, I ultimately believe it is not ready for publication in *Communications: Earth & Environment* in its current form. Several critical aspects need to be addressed to improve the overall transparency, reproducibility, and comprehensiveness of the study. To make the manuscript suitable for publication, I suggest the authors reconsider the following:

- Provide a more detailed explanation of the methodology and the use cases for a better overall understanding.
- Clarify the use and impact of historical adjustment factors, and consider including more direct comparisons with other models.
- Ensure greater transparency regarding the databases, inventories, and models (e.g., ecoinvent), and clearly state the contributions made by the authors beyond existing literature.

Communications Earth & Environment is committed to improving transparency in authorship. As part of our efforts in this direction, we are now requesting that all authors identified as 'corresponding author' create and link their Open Researcher and Contributor Identifier (ORCID) with their account on the Manuscript Tracking System prior to acceptance. ORCID helps the scientific community achieve unambiguous attribution of all scholarly contributions. You can create and link your ORCID from the home page of the Manuscript Tracking System by clicking on 'Modify my Springer Nature account' and following the instructions in the link below. Please also inform all co-authors that they can add their ORCIDs to their accounts and that

they must do so prior to acceptance.

Version 1:

Decision Letter:

<*** REMEMBER TO ATTACH REVISIONS CHECKLIST (WORD)***>

Dear Mr McFall,

Your manuscript titled "Air Travel Passenger Dynamic Emissions Calculator (**ATP-DEC**): A carbon footprint calculator for aviation with full LCA & historical adjustment factors" has now been seen by our reviewers, whose comments appear below. In light of their advice we are delighted to say that we are happy, in principle, to publish a suitably revised version in Communications Earth & Environment.

We therefore invite you to revise your paper one last time to address the remaining concerns of our reviewers. At the same time we ask that you edit your manuscript to comply with our format requirements and to maximise the accessibility and therefore the impact of your work.

EDITORIAL REQUESTS:

*****Please take care to match our formatting and policy requirements. We will check revised manuscript and return manuscripts that do not comply. Such requests will lead to delays. *****

SUBMISSION INFORMATION:

OPEN ACCESS:

Communications Earth & Environment is a fully open access journal. Articles are made freely accessible on publication. For further information about article processing charges, open access funding, and advice and support from Nature Portfolio, please visit <https://www.nature.com/commsenv/open-access>

Link Redacted

Best regards,

Nandita Basu, PhD

Consulting Editor, Communications Earth & Environment
Associate Editor, Communications Sustainability
Nature Portfolio

REVIEWERS' COMMENTS:

Reviewer #1 (Remarks to the Author):

Regarding the study titled "Air Travel Passenger Dynamic Emissions Calculator (ATP-DEC): A carbon footprint calculator for aviation with full LCA & historical adjustment factors", I saw that the issues I mentioned earlier were taken into consideration, the necessary corrections were made, and justifications were provided. I also read and reviewed that the authors took into account the comments made by the other referees and made the necessary corrections. I have no additional comments to make at this stage. I would like to thank everyone who contributed.

Reviewer #2 (Remarks to the Author):

Thank you for the author's revisions. The authors have addressed the issues mentioned earlier, and it is recommended that the manuscript be accepted.

Reviewer #3 (Remarks to the Author):

I would like to thank the authors for the detailed revision of the manuscript and their responses to the previous comments. The revised version represents a clear improvement, and most of my concerns have been addressed satisfactorily.

I have only a few minor remaining suggestions:

1. In the introduction, the last two paragraphs are somewhat redundant, as both outline the key questions and main focus of the tool. Additionally, it remains unclear who the primary users of the tool are: industry, policymakers, or the passengers. Clarifying the intended audience, even if multiple user groups are involved, would strengthen the introduction.
2. I appreciate that the authors have added a short section acknowledging sensitivity analysis (section 5.4.) as a limitation of the study. However, the text currently emphasizes the limitations of the Monte Carlo simulation, which may unintentionally suggest that this is the only available method for sensitivity analysis. It would be helpful to clarify that various other approaches to sensitivity analysis exist and could be considered in future work.
3. Finally, I recommend a consistency check throughout the manuscript, for example, regarding figure captions and the spelling of terms such as "lifecycle" vs. "life cycle".

With these minor revisions, I consider the manuscript suitable for publication

** Visit Nature Portfolio's author and referees' website at www.nature.com/authors for information about policies, services and author benefits**

REFeree REPORT

Referee: Dr. Selçuk Gürçam

Article Title: Air Travel Passenger Dynamic Emissions Calculator (ATP-DEC): A carbon footprint calculator for aviation with full LCA & historical adjustment factors

Author(s): F. McFall, D. Guan, J. Chenoweth, X. Font, I. Corduneanu, E. Goean, J. Sadhukhan

Publication Decision: Accepted – Minor Revision

General Comments

This study has proposed a new carbon footprint calculation methodology called ATP-DEC (Air Travel Passenger Dynamic Emissions Calculator), and is designed to compensate for the deficiencies in existing calculators, and is considered a valuable study in this respect. In fact, it is clear that the study offers a more realistic and innovative perspective on traditional flight-related carbon footprint measurement. However, despite its methodological depth and breadth, structural, conceptual, and analytical deficiencies are evident in certain aspects of the article. In this context, considering my field of study, the weaknesses that are particularly evident in the study are stated below:

First, it was seen that a clear research question or thesis statement was not clearly formulated in the study. Although the general purpose and areas of contribution are presented, a clearly defined research question that establishes a theoretical position and provides a basis for discussion is crucial in academic texts.

Second, the mathematical foundations of the calculation tools and algorithms used in the methodology section were not presented in detail. The significance of original contributions such as HAF (Historical Adjustment Factor) should be directly related to the technical transparency of these contributions. However, the study does not include detailed formulas, modeling processes, or data processing steps explaining how these calculations are performed. This situation is considered to limit the methodological reliability of the study.

Thirdly, the evaluations made on the existing literature generally remain at a descriptive level; a more in-depth critique can be provided in terms of the methodological origins, class assumptions or data infrastructures of the existing carbon calculators. I believe the literature review should not only summarize the tools and their functions but also evaluate their role and effects in the production of information.

Fourthly, although some incomplete calculations regarding engine efficiency, soot and sulfate aerosols are included in the 5.4 Model limitations and challenges section, the quantitative impact of these limitations on the estimated carbon emissions is not evaluated. This lack of evaluation may make it difficult for the reader to understand the potential deviation rate of the model.

Fifthly, in my opinion, the applicability analyses regarding how ATP-DEC will be used in practice are insufficient for policy makers, airline companies or individuals. For example, an empirical assessment can be made on how the working model will be applied in regions with low data infrastructure or to what extent it will meet user behaviors in developing countries. Issues such as in which contexts the model provides the most effective results, what infrastructure needs it requires, or how data sources change according to geographical differences can be addressed in depth.

Finally, I believe that the ATP-DEC methodology could make a significant contribution to the existing literature by providing a comprehensive, accurate, and transparent approach to calculating aviation emissions. After reviewing the working model by taking into account the issues I mentioned, I have no objection to its publication in terms of both academic studies and applications in the aviation sector.

The article presents an innovative aviation emissions calculation tool, ATP-DEC. Compared to existing carbon emission calculators, this tool has a broader scope of emissions calculation; it also introduces a historical adjustment factor (HAF) for dynamic adjustments based on historical flight data; adopts a comprehensive Life Cycle Assessment (LCA) method, considering the entire life cycle emissions of aircraft; and takes into account passenger cabin classes, luggage, and cargo, allowing for more detailed emission allocations for different passenger groups. Based on this research, I have the following questions that I hope the authors will address:

1. The article briefly discusses several existing tools (such as TIM, IATA, ICAO, MyClimate). It might be helpful to focus more on ATP-DEC and compare it within the context of existing research to strengthen the argument for ATP-DEC's innovation in this field.
2. Some assumptions made in the model (such as excluding certain non-Kyoto impact factors like soot and sulfate aerosols, and the selection of the historical adjustment factor (HAF)) lack sufficient justification. A more detailed discussion is needed on why these emission factors or effects were excluded, especially the potential role of soot and sulfate aerosols, which could impact the results.
3. Although the article mentions that ATP-DEC was validated against 30,000 historical flights, there is no discussion of the model's uncertainty. A more formal uncertainty analysis should be included, particularly given the complexity of dynamic modeling. A sensitivity analysis is recommended for future work.
4. The article mentions that ATP-DEC can be applied globally, but lacks an in-depth discussion on its applicability in different regions. The aviation operation modes, aircraft types, flight frequencies, and climate conditions in different regions could all affect the emission calculation results. It is suggested to add an analysis of the model's applicability in different regions and highlight potential limitations.
5. While the article compares ATP-DEC with existing tools, there is no direct comparison with international standards such as ISO 14083 or IPCC guidelines. It would be beneficial to include a comparison with these international standards in terms of methodology, emission factors, and data usage, demonstrating how ATP-DEC aligns with or exceeds these

industry standards.

6. The article primarily focuses on the analysis of existing flight data but lacks a predictive analysis of future aviation emission trends. For example, as the aviation industry transitions towards more environmentally friendly technologies (such as electric aircraft and sustainable aviation fuels), how will ATP-DEC adapt to these changes?
7. The article could also benefit from the inclusion of additional references to strengthen the background and contextualization of the research.

The article presents an innovative aviation emissions calculation tool, ATP-DEC. Compared to existing carbon emission calculators, this tool has a broader scope of emissions calculation; it also introduces a historical adjustment factor (HAF) for dynamic adjustments based on historical flight data; adopts a comprehensive Life Cycle Assessment (LCA) method, considering the entire life cycle emissions of aircraft; and takes into account passenger cabin classes, luggage, and cargo, allowing for more detailed emission allocations for different passenger groups. Based on this research, I have the following questions that I hope the authors will address:

1. The article briefly discusses several existing tools (such as TIM, IATA, ICAO, MyClimate). It might be helpful to focus more on ATP-DEC and compare it within the context of existing research to strengthen the argument for ATP-DEC's innovation in this field.
2. Some assumptions made in the model (such as excluding certain non-Kyoto impact factors like soot and sulfate aerosols, and the selection of the historical adjustment factor (HAF)) lack sufficient justification. A more detailed discussion is needed on why these emission factors or effects were excluded, especially the potential role of soot and sulfate aerosols, which could impact the results.
3. Although the article mentions that ATP-DEC was validated against 30,000 historical flights, there is no discussion of the model's uncertainty. A more formal uncertainty analysis should be included, particularly given the complexity of dynamic modeling. A sensitivity analysis is recommended for future work.
4. The article mentions that ATP-DEC can be applied globally, but lacks an in-depth discussion on its applicability in different regions. The aviation operation modes, aircraft types, flight frequencies, and climate conditions in different regions could all affect the emission calculation results. It is suggested to add an analysis of the model's applicability in different regions and highlight potential limitations.
5. While the article compares ATP-DEC with existing tools, there is no direct comparison with international standards such as ISO 14083 or IPCC guidelines. It would be beneficial to include a comparison with these international standards in terms of methodology, emission factors, and data usage, demonstrating how ATP-DEC aligns with or exceeds these

industry standards.

6. The article primarily focuses on the analysis of existing flight data but lacks a predictive analysis of future aviation emission trends. For example, as the aviation industry transitions towards more environmentally friendly technologies (such as electric aircraft and sustainable aviation fuels), how will ATP-DEC adapt to these changes?
7. The article could also benefit from the inclusion of additional references to strengthen the background and contextualization of the research.

The article presents an innovative aviation emissions calculation tool, ATP-DEC. Compared to existing carbon emission calculators, this tool has a broader scope of emissions calculation; it also introduces a historical adjustment factor (HAF) for dynamic adjustments based on historical flight data; adopts a comprehensive Life Cycle Assessment (LCA) method, considering the entire life cycle emissions of aircraft; and takes into account passenger cabin classes, luggage, and cargo, allowing for more detailed emission allocations for different passenger groups. Based on this research, I have the following questions that I hope the authors will address:

1. The article briefly discusses several existing tools (such as TIM, IATA, ICAO, MyClimate). It might be helpful to focus more on ATP-DEC and compare it within the context of existing research to strengthen the argument for ATP-DEC's innovation in this field.
2. Some assumptions made in the model (such as excluding certain non-Kyoto impact factors like soot and sulfate aerosols, and the selection of the historical adjustment factor (HAF)) lack sufficient justification. A more detailed discussion is needed on why these emission factors or effects were excluded, especially the potential role of soot and sulfate aerosols, which could impact the results.
3. Although the article mentions that ATP-DEC was validated against 30,000 historical flights, there is no discussion of the model's uncertainty. A more formal uncertainty analysis should be included, particularly given the complexity of dynamic modeling. A sensitivity analysis is recommended for future work.
4. The article mentions that ATP-DEC can be applied globally, but lacks an in-depth discussion on its applicability in different regions. The aviation operation modes, aircraft types, flight frequencies, and climate conditions in different regions could all affect the emission calculation results. It is suggested to add an analysis of the model's applicability in different regions and highlight potential limitations.
5. While the article compares ATP-DEC with existing tools, there is no direct comparison with international standards such as ISO 14083 or IPCC guidelines. It would be beneficial to include a comparison with these international standards in terms of methodology, emission factors, and data usage, demonstrating how ATP-DEC aligns with or exceeds these

industry standards.

6. The article primarily focuses on the analysis of existing flight data but lacks a predictive analysis of future aviation emission trends. For example, as the aviation industry transitions towards more environmentally friendly technologies (such as electric aircraft and sustainable aviation fuels), how will ATP-DEC adapt to these changes?
7. The article could also benefit from the inclusion of additional references to strengthen the background and contextualization of the research.

Responses to Reviewers

Air Travel Passenger Dynamic Emissions Calculator (ATP-DEC): A carbon footprint calculator for aviation with full LCA & historical adjustment factors

Reviewer 1 (Dr. Selçuk Gürçam)

- 1) “First, it was seen that a clear research question or thesis statement was not clearly formulated in the study. Although the general purpose and areas of contribution are presented, a clearly defined research question that establishes a theoretical position and provides a basis for discussion is crucial in academic texts”

We appreciate this feedback and have revised the introduction accordingly. In the revised manuscript, we have included a clearly stated research question at the end of the second paragraph in the introduction: “How a comprehensive carbon footprint calculator for aviation can be designed to offer maximum accuracy, transparency, and scope, while remaining practically usable by stakeholders with varying data availability.”

- 2) “Second, the mathematical foundations of the calculation tools and algorithms used in the methodology section were not presented in detail. The significance of original contributions such as HAF (Historical Adjustment Factor) should be directly related to the technical transparency of these contributions. However, the study does not include detailed formulas, modelling processes, or data processing steps explaining how these calculations are performed. This situation is considered to limit the methodological reliability of the study.”

We appreciate the comment and have revised the structure of the manuscript and supplementary information. Following editorial advice and the comments from all peer reviewers, we have moved the summarised step-by-step methodology to the main manuscript. This helps explain vital aspects of the ATP-DEC. In addition, a step-by-step explanation of an applied example is conducted and explained alongside the steps in the main manuscript, with cross referencing to the more in-depth supplementary method to improve readability and interconnectivity between the methodology and results. This includes a detailed HAF explanation of how its implemented and the processing steps in the method and in Section 5.2.

- 3) “Thirdly, the evaluations made on the existing literature generally remain at a descriptive level; a more in-depth critique can be provided in terms of the methodological origins, class assumptions or data infrastructures of the existing carbon calculators. I believe the literature review should not only summarize the tools and their functions but also evaluate their role and effects in the production of information.”

The Review of Different Tools section has been incorporated into the Introduction to make space for more important explanations in this article. The outcome of the review is included in the discussion with a comparison to ATP-DEC. The outcome is based on all the critique points mentioned in your thoughtful comment: methodological origins, class assumptions and data infrastructures. In addition, Table S1 and Table S3 in the Supplementary Information (SI) summarise the scope and accuracy of each calculator. We believe that our review of existing tools does evaluate their role and effects in the production of information by considering transparency, scope and effectiveness of communication.

- 4) “Fourthly, although some incomplete calculations regarding engine efficiency, soot and sulfate aerosols are included in the 5.4 Model limitations and challenges section, the

quantitative impact of these limitations on the estimated carbon emissions is not evaluated. This lack of evaluation may make it difficult for the reader to understand the potential deviation rate of the model.”

Thank you for this important observation. We have now revised Section 5.4 in the manuscript to provide a clearer discussion of this limitation.

While soot is a significant contributor to global warming, sulphate aerosols have a cooling effect, comparable in magnitude, helping to offset soot’s warming effect. Their modelling uncertainty has been recognised in the literature. Hence, an overall multiplier rather than individual aerosol components’ GWP characterisation factors has been considered.

The estimated mean error of the method for calculating non-Kyoto emissions has now been noted in section 5.4 and compared to the estimated mean error of applying a constant multiplier for non-Kyoto emissions.

- 5) “Fifthly, in my opinion, the applicability analyses regarding how ATP-DEC will be used in practice are insufficient for policy makers, airline companies or individuals. For example, an empirical assessment can be made on how the working model will be applied in regions with low data infrastructure or to what extent it will meet user behaviors in developing countries. Issues such as in which contexts the model provides the most effective results, what infrastructure needs it requires, or how data sources change according to geographical differences can be addressed in depth.”

Thank you for your comment. There are two points to your question:

First, you ask about the applicability of the tool for policy makers, airline companies and individuals. The purpose of our paper is to show the strengths of this methodology in providing more accurate and reliable data for the type of applications that the three commercially available tools are currently used for. While tool users may not vary per se, ATP-DEC provides users with more accurate and reliable calculations, increasing stakeholder trust. Discussing current and potential uses of carbon calculators is a lengthy process and many articles have been dedicated to it. We have therefore clarified the strengths of using a more rigorous tool, but we have deliberately stayed away from discussing the potential uses of that tool, beyond our explanation in the Conclusions section, in the paragraph starting with “Nudge to Environmental Action”. But we realise that some of the points explaining the potential usage of the tool were dispersed in different parts of the paper, and we have now summarised these at the end of the Introduction, to make them more obvious as the readers work their way through the paper. Within the word limits of the paper, we now summarise them as: “Compared to current tools in the market, ATP-DEC helps i) better raise awareness of the impact of flying and to attribute such impact to different sources, ii) more accurately track and report impacts for corporate reporting, iii) feed more accurate data to carbon offsetting mechanisms, iv) identify specific aspects of CO₂e that change over time, v) inform policies such as aviation taxation.”

Second, you ask us about the applicability of the model to regions with low data infrastructure. The redefinition of our research question now explicitly states “How a comprehensive carbon footprint calculator for air travel passengers can be designed to offer maximum accuracy, transparency, and scope, while remaining practically usable by stakeholders with varying data availability.”, hence the question you are asking yourself was at the centre of our methodological design. We acknowledge that the ATP-DEC is using a greater number of data points than other tools in the market (TTW, WTT, non-Kyoto impacts

(NO_x, H₂O, ClC), and lifecycle emissions from in-flight services, airports, and aircraft). This is following the principle established in ISO14040-44 and ISO14067, that all components of a product, in this case a flight, must be included for comprehensive quantification. Other tools use generalised weightings, while ATP-DEC has the option of using real data when this is available, or weighted data when it is not- this also encourages industry transparency. For example, both the luggage factor and cargo factor benefit from airline and route-specific passenger, luggage, and cargo weight data. We acknowledge that for some airlines this information will be unavailable, and for these cases the up-to-date 2022 review by EASA and Lufthansa is used to determine the mass of passengers and passenger luggage, depending on the plane capacity. We acknowledge that much of the data at airport level in developing countries will not be available, and yet the model shows how this data needs to be acknowledged but it does not make a material difference (e.g. figure 1), and following LCA methodology principles (ISO14040-44), sources accounting for less than 3% of the total can be estimated from weightings. In some instances, we use estimates from Ecoinvent, a globally recognised life cycle inventory (LCI) database that provides comprehensive, standardised, and peer-reviewed datasets essential for robust life cycle assessment (LCA). Much of the data that we have used is commercially available through sources like FlightRadar, SeatGuru and Eurocontrol. We acknowledge that we were not able to add an empirical assessment at this time, however the methodology in the online appendix shows the origin sources of our data and these are from official sources that are commercially available.

Reviewer 2

- 1) "The article briefly discusses several existing tools (such as TIM, IATA, ICAO, MyClimate). It might be helpful to focus more on ATP-DEC and compare it within the context of existing research to strengthen the argument for ATP-DEC's innovation in this field."

Thank you for your comment, it touched on an important point. Most of the content in the manuscript and SI focuses on the ATP-DEC and its benefits over existing tools. However, we have revised the structure of the manuscript and supplementary information. Following editorial advice and the comments from all peer reviewers, we have moved the step-by-step methodology to the main manuscript. This helps explain vital aspects of the ATP-DEC. In addition, a step-by-step explanation of an applied example is conducted in the manuscript, with cross referencing to improve readability and interconnectivity between the methodology, SI and results.

- 2) "Some assumptions made in the model (such as excluding certain non-Kyoto impact factors like soot and sulfate aerosols, and the selection of the historical adjustment factor (HAF)) lack sufficient justification. A more detailed discussion is needed on why these emission factors or effects were excluded, especially the potential role of soot and sulfate aerosols, which could impact the results."

While soot is a significant contributor to global warming, sulphate aerosols have a cooling effect, comparable in magnitude, helping to offset soot's warming effect. Their modelling uncertainty has been recognised in the literature. Hence, an overall multiplier rather than individual aerosol components' GWP characterisation factors has been considered.

The estimated mean error of the method for calculating non-Kyoto emissions has now been noted in section 5.4 and compared to the estimated mean error of applying a constant multiplier for non-Kyoto emissions.

Furthermore, a step-by-step explanation of an applied example of ATP-DEC is conducted in the manuscript, with cross referencing to improve readability and interconnectivity between the methodology and results. This includes a detailed HAF explanation of how its implemented and the processing steps. The HAF validation is further explained in Section 5.2.

- 3) “Although the article mentions that ATP-DEC was validated against 30,000 historical flights, there is no discussion of the model’s uncertainty. A more formal uncertainty analysis should be included, particularly given the complexity of dynamic modeling. A sensitivity analysis is recommended for future work.”

We appreciate this comment, and we agree that a sensitivity analysis, such as a Monte Carlo simulation, would be very helpful to show the spread/standard deviation of GWP due to sensitive variables. Uncertainty quantification is beyond the current scope of this article (due to both word count and timeliness of the research), but we have acknowledged in the text (Section 5.4) that we plan to conduct a comprehensive sensitivity analysis as part of future work. Specifically, we’re aware of the emergence of SAF, so we are hoping to do a dedicated example of how SAF emission factors will alter the output of ATP-DEC in future work.

- 4) “The article mentions that ATP-DEC can be applied globally, but lacks an in-depth discussion on its applicability in different regions. The aviation operation modes, aircraft types, flight frequencies, and climate conditions in different regions could all affect the emission calculation results. It is suggested to add an analysis of the model’s applicability in different regions and highlight potential limitations.”

Thank you for this comment and please see comment 5 from reviewer 1 (above) as our answer overlaps.

To specifically address your queried variables:

- Operation modes / aircraft type: ATP-DEC is flexible with aircraft type if fuel data is available. Also, the aircraft age variable considers engine efficiency.
- Flight frequencies: ATP-DEC is not constrained by frequency of flight.
- Climate conditions: modelling climate conditions for a passenger air travel calculator is very difficult, but the HAF aims to track patterns in flight paths. Therefore, consistent weather conditions should be accounted for through HAF.
- ATP-DEC is intended to be globally applicable as nearly all commercial flights are tracked and monitored.

- 5) “While the article compares ATP-DEC with existing tools, there is no direct comparison with international standards such as ISO 14083 or IPCC guidelines. It would be beneficial to include a comparison with these international standards in terms of methodology, emission factors, and data usage, demonstrating how ATP-DEC aligns with or exceeds these industry standards.”

ISO 14083 is now consistently mentioned throughout the revised manuscript. We aim to link it within the methodology to give readers the understanding that some parts of the method align with the standard and some parts go beyond the standard. This is further demonstrated in the worked example.

- 6) The article primarily focuses on the analysis of existing flight data but lacks a predictive analysis of future aviation emission trends. For example, as the aviation industry

transitions towards more environmentally friendly technologies (such as electric aircraft and sustainable aviation fuels), how will ATP-DEC adapt to these changes?"

ATP-DEC is flexible to fuel type by having a customisable fuel emission factor variable. The default value used is Jet A1 fuel, but this can be altered when SAF or electric aircraft are used in the future (both still have an emission factor greater than zero). ATP-DEC is designed to adapt through the modular design, which is emphasised in the revised statement: "Furthermore, ATP-DEC is modular so it's flexible to evolving science and data. Each individual constituent to ATP-DEC can be reworked without affecting the others. This also encourages advancement. For example, the default value for fuel emission factors is Jet A1 fuel. But this can be easily altered when SAF or electric aircraft are used in the future".

Whilst ATP-DEC does not currently support predictive analysis in the form of ML/AI, the HAF uses historic data to improve future predictions. ATP-DEC of course is meant to be used as a predictive tool, to be used when booking future flights.

7) "The article could also benefit from the inclusion of additional references to strengthen the background and contextualization of the research."

We appreciate this comment, and several references have been added to the main manuscript as well as the SI. As well as some of our own, please see comment 4 by reviewer 3.

Reviewer 3

1) "The abbreviations "MSPE" and "ATP-DEC" are used in the abstract without being introduced. It would be helpful to define them upon first use for better clarity."

We appreciate this comment picking up on this. ATP-DEC is defined in the title of the article. Meanwhile, MSPE has now been defined in the abstract, where it is first used.

2) "The terms "life cycle assessment" and "carbon footprint" are used somewhat interchangeably. However, LCA is a standardized, comprehensive method that evaluates various environmental impacts across multiple impact categories, while a carbon footprint specifically focuses on greenhouse gas emissions throughout a product's life cycle."

Thank you for this comment. We have worked through the manuscript and SI and made sure that the use of these terms is consistent. "Life cycle assessment" or "LCA" is used when discussing the approach, which is now clearly defined (see comment 16 of this review). "Carbon footprint" or "emissions footprint" is used to describe what we are calculating/estimating, with GWP per passenger as the metric.

3) "The term "non-Kyoto emissions" could be confusing, as it implies a different context. I recommend the authors clarify this terminology or use the more widely accepted "non-CO₂ emissions" instead."

We appreciate this thoughtful suggestion. However, we have intentionally retained the term "non-Kyoto emissions" to distinguish between greenhouse gases regulated under the Kyoto Protocol and other aviation-specific climate forcers.

While the term "non-CO₂ emissions" is widely used, in the context of aviation it can lead to ambiguity because it technically excludes non-CO₂ gases within the Kyoto protocol (e.g.,

CH₄ and N₂O). These emissions are already accounted for separately within our LCA framework. Our use of “non-Kyoto emissions” refers specifically to climate forcers not included in the Kyoto Protocol, such as NO_x, H₂O, and contrail-induced cirrus effects.

Given that these are evaluated using global warming potential (GWP) and expressed in CO₂-equivalent (CO₂e) terms we believe “non-Kyoto emissions” offers the most precise and transparent terminology for this context. To reduce any potential for confusion, we have refined the terminology by saying “non-Kyoto impacts, a term we use to refer to substances created by aeroplanes not covered under the Kyoto Protocol, such as nitrogen oxide (NO_x), water vapor (H₂O), and contrail induced cloudiness (CiC)”. This clarifies that we are not attempting to coin a new term but improve transparency within our study.

- 4) “The literature review is very short, and the gaps in existing research are not sufficiently highlighted. It would strengthen the paper if the authors could provide a more thorough review of relevant literature and more clearly articulate the gaps and their implications for the field. Current relevant papers in this field include:
- Rupcic et al. (2023): Environmental impacts in the civil aviation sector: Current state and guidance. Doi:10.1016/j.trd.2023.103717
 - Cox et al. (2018): Life cycle assessment of air transportation and the Swiss commercial air transport fleet. Doi: 10.1016/j.trd.2017.10.017
 - Rahn et al. (2025): Quantifying climate impacts of flight operations: A discrete-event life cycle assessment approach. Doi: 10.1016/j.trd.2025.104646
 - Keiser et al. (2023): Life cycle assessment in aviation: A systematic literature review of applications, methodological approaches and challenges. Doi: 10.1016/j.jairtraman.2023.102418”

Thank you for these insightful references. They have been included in the manuscript and SI:

- Rupcic et al. reinforces the introduction in the main manuscript in and the goal and scope definition section in SI.
- Cox et al. provides evidence for the importance of upstream emissions, referenced in the Review section in the main manuscript.
- Rahn et al. emphasises the importance of including non-Kyoto impacts and operational scenarios (HAF) into LCA. This is referenced in the literature review and discussion.
- Keiser et al. reinforces the arguments for the importance of a comprehensive LCA-based approach.

- 5) “The statement “ The non-Kyoto effects of aviation on climate are substantially greater than the CO₂ effects” in the beginning of chapter 3 is very vague. I recommend providing quantitative data or more specific examples.”

This comment is appreciated, and we have revised the statement to: “The non-Kyoto effects of aviation on climate can be substantially greater than the CO₂ effects” and cited Lee et al.¹.

- 6) “The manuscript emphasizes the necessity of a full cradle-to-grave model to assess the environmental footprint of flights. However, the rationale for this approach is not clear. The authors should clarify what “cradle” and “grave” refer to in the context of aviation and why this model is vital for the assessment.”

We have defined a subsection in the methodology section, now within the main manuscript, to clarify the system scope and the rationale for the full cradle to grave approach in the context of the ATP-DEC.

“Cradle” = Everything from raw-material extraction (mining, refining, composite production) through aircraft assembly and airport/infrastructure construction up to “first flight.”

“Grave” = Everything from aircraft retirement (disassembly, recycling, waste handling) through decommissioning or repurposing of airport facilities.

Why is this vital? Because aviation’s total environmental footprint spans far beyond the fuel uses. A full cradle-to-grave lifecycle model as recommended by ISO 14083 captures hidden CO₂, pollutants, resource depletion, and waste streams that an “in-flight-only” analysis would completely miss and therefore underestimate. By bringing out the life cycle systemic carbon footprint ², actionable offsetting mechanisms ³ can be developed. The cradle-to-grave system scope in aviation sector’s context includes the production and transportation of fuel (Well-To-Tank (WTT)), fuel burning during flying, take-off and landing (Tank-To-Wake (TTW)), in-flight services, and airport and aircraft life cycles. As can be seen in Figure 1, TTW mainly contributed by fuel burning, is a small contributor to the overall GWP. The system consideration is the most comprehensive far beyond the capability of the current aviation sector’s carbon footprint calculator.

7) “The challenges in integrating non-CO₂ effects into assessments are only briefly mentioned. A more thorough literature review on why these effects are difficult to quantify (e.g., due to altitude, location, or other factors) would significantly improve the study's context and depth.”

We have explained the importance of non-Kyoto, its definition, and why it is currently left out of existing tools. This has been addressed in the introduction.

To clarify:

- Non-Kyoto effects are sensitive to atmospheric conditions, altitude, latitude, time of day/season and climate. Also, some impacts have short term climate effects and only persist for hours. For this reason, ATR is preferred as a metric.
- Computationally expensive tools exist that can precisely calculate the contrail and non-CO₂ emissions of historical flights ⁴. However, directly calculating contrail impact for each flight would require complex climate modelling and is unfeasible.
- ATP-DEC address these challenges by using the most advanced available simplified model developed by Dahlmann et al. ⁵. We acknowledge the uncertainty in Dahlmann et al. To help combat one of the uncertainties arising from the use of Dahlmann et al.’s approach, the HAF method considers mean latitude and distance of each flight.
- The non-Kyoto impact is an additional component that can be toggled or updated as science improves. This is another selling point for the modular breakdown design of ATP-DEC.

8) “The content and purpose of figure 1 is not completely clear to me. What is the idea behind this figure? The assumption of an aircraft with only first-class seats is unrealistic and I don’t see the added-value in this graphic.”

This Figure may have been misinterpreted. Figure 1 shows the emissions footprint breakdown for a flight for four existing calculators and ATP-DEC. It includes a comparison of First, Business, Premium Economy, and Economy seats. The figure does not assume an

aircraft of only first-class seats. Now this figure may be more clear with the explanation of how the results are produced in manuscript method.

- 9) “The authors mention the often-overlooked emissions from luggage transportation. However, isn’t this typically included within passenger-related emissions (PAX)? Clarification on this point would strengthen the argument.”

Thank you for your comment and you are correct: luggage transportation is often overlooked. Existing calculators lump a standard luggage allowance into the per-passenger emission calculation. ATP-DEC separates passenger body weight and luggage weight and allocates emissions accordingly. This means in the implementation of ATP-DEC, when a user decides to take more luggage with them (whether its hold luggage or hand luggage), they can add this separately to their weight. This is added value for the ATP-DEC and is now more clearly explained in the methodology found in the main manuscript within the worked example. Table S2 now shows the data behind the luggage factor.

- 10) “The tools mentioned are difficult for comparison as they do not include non-CO₂ effects. The authors should discuss this limitation more explicitly and explain how their approach provides a more comprehensive assessment.”

Thank you for this comment and we understand the issue. One goal of this article is to encourage change in existing methodologies to include non-Kyoto impacts. We therefore want to make apparent the vast difference between the tools with and without these impacts. This limitation is discussed at length in the comparisons and results.

Existing tools inherently report lower emissions for a given flight in many cases because, as you correctly point out, they often omit non-Kyoto impacts. This is why, for comparison, we have provided a full breakdown of the ATP-DEC output in Figure 1 (this also exemplifies another strength of the ATP-DEC). Readers can understand the impact with and without the non-Kyoto emissions.

Furthermore, when we conduct a direct comparison with the TIM-based model, the non-Kyoto emissions were removed from the ATP-DEC output to allow fair comparison.

- 11) “The structure of the use cases is unclear. The inclusion of different flights and time frames (e.g., pre-pandemic and pre-Russian occupation periods) lacks organization and makes the methodology hard to follow. A dedicated chapter explaining the methodology and choices made for these use cases/comparisons would greatly enhance clarity.”

The revised SI explains the motivation behind the choice of flight operations data in Section 1.4. To validate the ATP-DEC (specifically the HAF algorithm), historical flight data was acquired commercially through FlightRadar24⁶. To test the HAF algorithm we searched for two flight operation databases either side of a global event, for which we selected the closure of Russian airspace for most airlines in early 2022. The data was therefore taken from 2019 and 2023 to avoid the disruption of the pandemic. Six flight routes were selected because they fly directly over Russia⁷.

- 12) “The purpose and relevance of the rolling window length (chapter 4.4) are not clearly explained.”

Thank you for your comment. For the HAF, a shorter rolling window might capture recent trends and anomalies but could be noisy, whereas a longer window smooths out fluctuations but might be slower to respond to sudden changes. This section is an investigation and informs the decision of which window to use (by default or by the user) in ATP-DEC. In

Section 5.2 we state on revision that 7-day window is adopted as default for the model going forward. This gives the more purpose and relevance to the rolling window length.

13) "What does DOI in table 2 mean?"

Thank you for your comment. This was a misunderstanding on the authors part and has now been corrected to "Accessed: dd mm yy", meaning the date of access when we conducted the research.

14) "The methodology behind the tool is not sufficiently explained. How does it work and how can a user work apply it? A detailed description of how the tool operates and its underlying assumptions is crucial for understanding and evaluating its validity."

The revised method section in the main manuscript clarifies how the tool works. We have included a worked example for additional understanding and to link the method with the results.

15) "The manuscript does not allow for reproducibility of the results, which is a fundamental aspect of scientific research. Clear documentation of methods and data sources is needed to address this issue. Additionally, the manuscript does not specify which database version or datasets were used. Including these details is essential for transparency and verification of the results."

We appreciate the comment and have revised the structure of the manuscript and supplementary information. Following editorial advice and the comments from all peer reviewers, we have moved the fine-tuned methodology to the main manuscript. This helps explain vital aspects of the ATP-DEC. In addition, a step-by-step explanation of an applied example is conducted in the SI, with cross referencing to improve readability and interconnectivity between the methodology and results.

In terms of documentation of data sources, we have ensured that all data sources are referenced in the manuscript and SI.

16) "While the manuscript claims to consider "full life flight operations," it does not follow the standardized LCA methodology (based on ISO 14040/44). Key elements such as the definition of goals, scope, and inventory creation are unclear, which undermines the transparency of the study."

We appreciate the comment. The main manuscript now contains the revised methodology with the structure more clearly following the standardised ISO 14040/44 LCA methodology. This is achieved by a designated Goal and Scope Definition section and the basis for Inventory Analysis and Impact Assessment. The former clarifies the system boundary and scopes as well as objectives of the study. The latter is equation or formulation driven and has been detailed through the equations for each life cycle activity. The interpretation shows the step-by-step calculations to use ATP-DEC and generate the results, for example, as those illustrated in the main paper. The transparency has been improved through illustrative examples for each formulation. This can be found in the method in the main manuscript, where the application example from Figure 1 is explained step by step, each equation substituted with the parametric and variable values.

17) "The inclusion of non-CO₂ effects is presented as a unique selling point. However, the equations used appear to be entirely adapted from Dahlmann et al. (2023), where specific assumptions for their validity are explicitly stated. It is unclear whether these assumptions are actually met in the context of this manuscript. The authors should

carefully evaluate whether the assumptions underlying these equations are satisfied and clearly justify their applicability to this study”

In section 5.4 an additional model limitation has been noted about the Dahlmann et al method and its assumptions. The estimated mean error of the method for calculating non-Kyoto emissions has now been noted and compared to the estimated mean error of applying a constant multiplier for non-Kyoto emissions.

18) “The supplementary material is very detailed, organized, and useful, but it is difficult to fully understand the paper without any explanation of the aspects presented in it. It would be helpful to clarify how and why these elements are relevant to the study within the main text, as this would improve the overall readability and comprehension of the paper.”

We appreciate the comment and have revised the structure of the manuscript and supplementary information. Following editorial advice and the comments from all peer reviewers, we have moved the fine-tuned methodology to the main manuscript. This helps explain vital aspects of the ATP-DEC. In addition, a step-by-step explanation of an applied example is conducted in the SI, with cross referencing to improve readability and interconnectivity between the methodology and results.

Reviewer 3 Summary

19) “Provide a more detailed explanation of the methodology and the use cases for a better overall understanding.”

Thank you for your detailed comments on the methodology. We believe we have understood your viewpoint and improved the explanation of the methodology substantially in the manuscript and SI. The manuscript now clearly explains the step-by-step method with the outline of the example calculation. The SI now goes through the example in detail. Throughout the text the figures and equations are referenced for clarity.

20) “Clarify the use and impact of historical adjustment factors, and consider including more direct comparisons with other models.”

Thank you for taking the time to understand the HAF. We have clarified the use and impact of HAF through a detailed explanation of the example calculation with an example equation. Figures 1, 2, 3, 4, and 5 all show the impact of the HAF and Figures 1 and 2 show direct comparison with other models.

21) “Ensure greater transparency regarding the databases, inventories, and models (e.g.,ecoinvent), and clearly state the contributions made by the authors beyond existing literature.”

We appreciate the comments on transparency, since this is one of the main goals of the ATP-DEC: full transparency. We have referenced all databases used, in both the main manuscript and the SI. All data used in the results has been referenced and some has been added in tables in the SI for full reproducibility.

Each author has made a substantial contribution to the concept and design of the work. The contributions of the authors include:

F McFall – writing, editing, concept, interpretation of data, creation of software

D Guan – editing, interpretation of data, acquisition of data

J Chenoweth – writing, concept, background research

X Font – writing, concept, background research, policy implementation

I Corduneanu – concept, editing, interpretation of data, acquisition of data

E Goean – concept, editing, interpretation of data acquisition of data

J Sadhukhan – concept, writing, editing, interpretation of data

References

- 1 Lee, D. S. *et al.* The contribution of global aviation to anthropogenic climate forcing for 2000 to 2018. *Atmospheric environment* **244**, 117834 (2021).
- 2 Sadhukhan, J. Net zero electricity systems in global economies by life cycle assessment (LCA) considering ecosystem, health, monetization, and soil CO₂ sequestration impacts. *Renewable Energy* **184**, 960-974 (2022).
- 3 Goean, E. R. *et al.* Using the Blockchain to Reduce Carbon Emissions in the Visitor Economy. *Sustainability* **16**, 4000 (2024).
- 4 Estuaire. Estuaire: Climate analytics for aviation. *Estuaire* (2024).
- 5 Dahlmann, K., Grewe, V., Matthes, S. & Yamashita, H. Climate assessment of single flights: Deduction of route specific equivalent CO₂ emissions. *International Journal of Sustainable Transportation* **17**, 29-40 (2023).
- 6 FlightRadar, A. B. FlightRadar24: Live Flight Tracker. *FlightRadar24* (2025).
- 7 Chu, C., Zhang, H., Zhang, J., Cong, L. & Lu, F. Assessing impacts of the Russia-Ukraine conflict on global air transportation: From the view of mass flight trajectories. *Journal of Air Transport Management* **115**, 102522 (2024).

Responses to Reviewers

Aviation passenger carbon footprint calculator with comprehensive emissions, life cycle coverage, and historical adjustment

Second Resubmission 26/06/25

Reviewer 1

“Regarding the study titled "Air Travel Passenger Dynamic Emissions Calculator (ATP-DEC): A carbon footprint calculator for aviation with full LCA & historical adjustment factors", I saw that the issues I mentioned earlier were taken into consideration, the necessary corrections were made, and justifications were provided. I also read and reviewed that the authors took into account the comments made by the other referees and made the necessary corrections. I have no additional comments to make at this stage. I would like to thank everyone who contributed.”

We thank the reviewer for their positive feedback and for acknowledging the revisions made in response to all reviewers' comments. We appreciate their time and constructive input throughout the review process.

Reviewer 2

“Thank you for the author's revisions. The authors have addressed the issues mentioned earlier, and it is recommended that the manuscript be accepted.”

We thank the reviewer for their positive feedback and for acknowledging the revisions made in response to all reviewers' comments. We appreciate their time and constructive input throughout the review process.

Reviewer 3

- 1) “In the introduction, the last two paragraphs are somewhat redundant, as both outline the key questions and main focus of the tool. Additionally, it remains unclear who the primary users of the tool are: industry, policymakers, or the passengers. Clarifying the intended audience, even if multiple user groups are involved, would strengthen the introduction.

We thank the reviewer for this comment. In the revised manuscript, we have combined the final two paragraphs of the introduction to improve coherence. We have also clarified that the primary intended users of the tool are stakeholders within the aviation industry.

- 2) “I appreciate that the authors have added a short section acknowledging sensitivity analysis (section 5.4.) as a limitation of the study. However, the text currently emphasizes the limitations of the Monte Carlo simulation, which may unintentionally suggest that this is the only available method for sensitivity analysis. It would be helpful to clarify that various other approaches to sensitivity analysis exist and could be considered in future work.

In the revised manuscript, Section 5.4 has been updated to note that various approaches to sensitivity analysis exist, including both deterministic and probabilistic single- or multi-variable methods. We also provide examples of these.

- 3) “Finally, I recommend a consistency check throughout the manuscript, for example, regarding figure captions and the spelling of terms such as "lifecycle" vs. "life cycle”.

We thank the reviewer for the recommendation. We have carried out a thorough consistency check throughout the manuscript, including figure captions, and ensured consistent terminology, using “life cycle” as two separate words.

First Resubmission 26/06/25

Reviewer 1 (Dr. Selçuk Gürçam)

- 1) “First, it was seen that a clear research question or thesis statement was not clearly formulated in the study. Although the general purpose and areas of contribution are presented, a clearly defined research question that establishes a theoretical position and provides a basis for discussion is crucial in academic texts”

We appreciate this feedback and have revised the introduction accordingly. In the revised manuscript, we have included a clearly stated research question at the end of the second paragraph in the introduction: “How a comprehensive carbon footprint calculator for aviation can be designed to offer maximum accuracy, transparency, and scope, while remaining practically usable by stakeholders with varying data availability.”

- 2) “Second, the mathematical foundations of the calculation tools and algorithms used in the methodology section were not presented in detail. The significance of original contributions such as HAF (Historical Adjustment Factor) should be directly related to the technical transparency of these contributions. However, the study does not include detailed formulas, modelling processes, or data processing steps explaining how these calculations are performed. This situation is considered to limit the methodological reliability of the study.”

We appreciate the comment and have revised the structure of the manuscript and supplementary information. Following editorial advice and the comments from all peer reviewers, we have moved the summarised step-by-step methodology to the main manuscript. This helps explain vital aspects of the ATP-DEC. In addition, a step-by-step explanation of an applied example is conducted and explained alongside the steps in the main manuscript, with cross referencing to the more in-depth supplementary method to improve readability and interconnectivity between the methodology and results. This includes a detailed HAF explanation of how its implemented and the processing steps in the method and in Section 5.2.

- 3) “Thirdly, the evaluations made on the existing literature generally remain at a descriptive level; a more in-depth critique can be provided in terms of the methodological origins, class assumptions or data infrastructures of the existing carbon calculators. I believe the literature review should not only summarize the tools and their functions but also evaluate their role and effects in the production of information.”

The Review of Different Tools section has been incorporated into the Introduction to make space for more important explanations in this article. The outcome of the review is included in the discussion with a comparison to ATP-DEC. The outcome is based on all the critique points mentioned in your thoughtful comment: methodological origins, class assumptions and data infrastructures. In addition, Table S1 and Table S3 in the Supplementary Information (SI) summarise the scope and accuracy of each calculator. We believe that our review of existing tools does evaluate their role and effects in the production of information by considering transparency, scope and effectiveness of communication.

- 4) “Fourthly, although some incomplete calculations regarding engine efficiency, soot and sulfate aerosols are included in the 5.4 Model limitations and challenges section, the quantitative impact of these limitations on the estimated carbon emissions is not evaluated. This lack of evaluation may make it difficult for the reader to understand the potential deviation rate of the model.”

Thank you for this important observation. We have now revised Section 5.4 in the manuscript to provide a clearer discussion of this limitation.

While soot is a significant contributor to global warming, sulphate aerosols have a cooling effect, comparable in magnitude, helping to offset soot's warming effect. Their modelling uncertainty has been recognised in the literature. Hence, an overall multiplier rather than individual aerosol components' GWP characterisation factors has been considered.

The estimated mean error of the method for calculating non-Kyoto emissions has now been noted in section 5.4 and compared to the estimated mean error of applying a constant multiplier for non-Kyoto emissions.

- 5) "Fifthly, in my opinion, the applicability analyses regarding how ATP-DEC will be used in practice are insufficient for policy makers, airline companies or individuals. For example, an empirical assessment can be made on how the working model will be applied in regions with low data infrastructure or to what extent it will meet user behaviors in developing countries. Issues such as in which contexts the model provides the most effective results, what infrastructure needs it requires, or how data sources change according to geographical differences can be addressed in depth."

Thank you for your comment. There are two points to your question:

First, you ask about the applicability of the tool for policy makers, airline companies and individuals. The purpose of our paper is to show the strengths of this methodology in providing more accurate and reliable data for the type of applications that the three commercially available tools are currently used for. While tool users may not vary per se, ATP-DEC provides users with more accurate and reliable calculations, increasing stakeholder trust. Discussing current and potential uses of carbon calculators is a lengthy process and many articles have been dedicated to it. We have therefore clarified the strengths of using a more rigorous tool, but we have deliberately stayed away from discussing the potential uses of that tool, beyond our explanation in the Conclusions section, in the paragraph starting with "Nudge to Environmental Action". But we realise that some of the points explaining the potential usage of the tool were dispersed in different parts of the paper, and we have now summarised these at the end of the Introduction, to make them more obvious as the readers work their way through the paper. Within the word limits of the paper, we now summarise them as: "Compared to current tools in the market, ATP-DEC helps i) better raise awareness of the impact of flying and to attribute such impact to different sources, ii) more accurately track and report impacts for corporate reporting, iii) feed more accurate data to carbon offsetting mechanisms, iv) identify specific aspects of CO₂e that change over time, v) inform policies such as aviation taxation."

Second, you ask us about the applicability of the model to regions with low data infrastructure. The redefinition of our research question now explicitly states "How a comprehensive carbon footprint calculator for air travel passengers can be designed to offer maximum accuracy, transparency, and scope, while remaining practically usable by stakeholders with varying data availability.", hence the question you are asking yourself was at the centre of our methodological design. We acknowledge that the ATP-DEC is using a greater number of data points than other tools in the market (TTW, WTT, non-Kyoto impacts (NO_x, H₂O, ClC), and lifecycle emissions from in-flight services, airports, and aircraft). This is following the principle established in ISO14040-44 and ISO14067, that all components of a product, in this case a flight, must be included for comprehensive quantification. Other tools use generalised weightings, while ATP-DEC has the option of using real data when this is

available, or weighted data when it is not- this also encourages industry transparency. For example, both the luggage factor and cargo factor benefit from airline and route-specific passenger, luggage, and cargo weight data. We acknowledge that for some airlines this information will be unavailable, and for these cases the up-to-date 2022 review by EASA and Lufthansa is used to determine the mass of passengers and passenger luggage, depending on the plane capacity. We acknowledge that much of the data at airport level in developing countries will not be available, and yet the model shows how this data needs to be acknowledged but it does not make a material difference (e.g. figure 1), and following LCA methodology principles (ISO14040-44), sources accounting for less than 3% of the total can be estimated from weightings. In some instances, we use estimates from Ecoinvent, a globally recognised life cycle inventory (LCI) database that provides comprehensive, standardised, and peer-reviewed datasets essential for robust life cycle assessment (LCA). Much of the data that we have used is commercially available through sources like FlightRadar, SeatGuru and Eurocontrol. We acknowledge that we were not able to add an empirical assessment at this time, however the methodology in the online appendix shows the origin sources of our data and these are from official sources that are commercially available.

Reviewer 2

- 1) "The article briefly discusses several existing tools (such as TIM, IATA, ICAO, MyClimate). It might be helpful to focus more on ATP-DEC and compare it within the context of existing research to strengthen the argument for ATP-DEC's innovation in this field."

Thank you for your comment, it touched on an important point. Most of the content in the manuscript and SI focuses on the ATP-DEC and its benefits over existing tools. However, we have revised the structure of the manuscript and supplementary information. Following editorial advice and the comments from all peer reviewers, we have moved the step-by-step methodology to the main manuscript. This helps explain vital aspects of the ATP-DEC. In addition, a step-by-step explanation of an applied example is conducted in the manuscript, with cross referencing to improve readability and interconnectivity between the methodology, SI and results.

- 2) "Some assumptions made in the model (such as excluding certain non-Kyoto impact factors like soot and sulfate aerosols, and the selection of the historical adjustment factor (HAF)) lack sufficient justification. A more detailed discussion is needed on why these emission factors or effects were excluded, especially the potential role of soot and sulfate aerosols, which could impact the results."

While soot is a significant contributor to global warming, sulphate aerosols have a cooling effect, comparable in magnitude, helping to offset soot's warming effect. Their modelling uncertainty has been recognised in the literature. Hence, an overall multiplier rather than individual aerosol components' GWP characterisation factors has been considered.

The estimated mean error of the method for calculating non-Kyoto emissions has now been noted in section 5.4 and compared to the estimated mean error of applying a constant multiplier for non-Kyoto emissions.

Furthermore, a step-by-step explanation of an applied example of ATP-DEC is conducted in the manuscript, with cross referencing to improve readability and interconnectivity between the methodology and results. This includes a detailed HAF explanation of how it is implemented and the processing steps. The HAF validation is further explained in Section 5.2.

- 3) “Although the article mentions that ATP-DEC was validated against 30,000 historical flights, there is no discussion of the model’s uncertainty. A more formal uncertainty analysis should be included, particularly given the complexity of dynamic modeling. A sensitivity analysis is recommended for future work.”

We appreciate this comment, and we agree that a sensitivity analysis, such as a Monte Carlo simulation, would be very helpful to show the spread/standard deviation of GWP due to sensitive variables. Uncertainty quantification is beyond the current scope of this article (due to both word count and timeliness of the research), but we have acknowledged in the text (Section 5.4) that we plan to conduct a comprehensive sensitivity analysis as part of future work. Specifically, we’re aware of the emergence of SAF, so we are hoping to do a dedicated example of how SAF emission factors will alter the output of ATP-DEC in future work.

- 4) “The article mentions that ATP-DEC can be applied globally, but lacks an in-depth discussion on its applicability in different regions. The aviation operation modes, aircraft types, flight frequencies, and climate conditions in different regions could all affect the emission calculation results. It is suggested to add an analysis of the model’s applicability in different regions and highlight potential limitations.”

Thank you for this comment and please see comment 5 from reviewer 1 (above) as our answer overlaps.

To specifically address your queried variables:

- Operation modes / aircraft type: ATP-DEC is flexible with aircraft type if fuel data is available. Also, the aircraft age variable considers engine efficiency.
- Flight frequencies: ATP-DEC is not constrained by frequency of flight.
- Climate conditions: modelling climate conditions for a passenger air travel calculator is very difficult, but the HAF aims to track patterns in flight paths. Therefore, consistent weather conditions should be accounted for through HAF.
- ATP-DEC is intended to be globally applicable as nearly all commercial flights are tracked and monitored.

- 5) “While the article compares ATP-DEC with existing tools, there is no direct comparison with international standards such as ISO 14083 or IPCC guidelines. It would be beneficial to include a comparison with these international standards in terms of methodology, emission factors, and data usage, demonstrating how ATP-DEC aligns with or exceeds these industry standards.”

ISO 14083 is now consistently mentioned throughout the revised manuscript. We aim to link it within the methodology to give readers the understanding that some parts of the method align with the standard and some parts go beyond the standard. This is further demonstrated in the worked example.

- 6) The article primarily focuses on the analysis of existing flight data but lacks a predictive analysis of future aviation emission trends. For example, as the aviation industry transitions towards more environmentally friendly technologies (such as electric aircraft and sustainable aviation fuels), how will ATP-DEC adapt to these changes?”

ATP-DEC is flexible to fuel type by having a customisable fuel emission factor variable. The default value used is Jet A1 fuel, but this can be altered when SAF or electric aircraft are used in the future (both still have an emission factor greater than zero). ATP-DEC is designed to adapt through the modular design, which is emphasised in the revised

statement: “Furthermore, ATP-DEC is modular so it’s flexible to evolving science and data. Each individual constituent to ATP-DEC can be reworked without affecting the others. This also encourages advancement. For example, the default value for fuel emission factors is Jet A1 fuel. But this can be easily altered when SAF or electric aircraft are used in the future”.

Whilst ATP-DEC does not currently support predictive analysis in the form of ML/AI, the HAF uses historic data to improve future predictions. ATP-DEC of course is meant to be used as a predictive tool, to be used when booking future flights.

7) “The article could also benefit from the inclusion of additional references to strengthen the background and contextualization of the research.”

We appreciate this comment, and several references have been added to the main manuscript as well as the SI. As well as some of our own, please see comment 4 by reviewer 3.

Reviewer 3

1) “The abbreviations “MSPE” and “ATP-DEC” are used in the abstract without being introduced. It would be helpful to define them upon first use for better clarity.”

We appreciate this comment picking up on this. ATP-DEC is defined in the title of the article. Meanwhile, MSPE has now been defined in the abstract, where it is first used.

2) “The terms “life cycle assessment” and “carbon footprint” are used somewhat interchangeably. However, LCA is a standardized, comprehensive method that evaluates various environmental impacts across multiple impact categories, while a carbon footprint specifically focuses on greenhouse gas emissions throughout a product’s life cycle.”

Thank you for this comment. We have worked through the manuscript and SI and made sure that the use of these terms is consistent. “Life cycle assessment” or “LCA” is used when discussing the approach, which is now clearly defined (see comment 16 of this review). “Carbon footprint” or “emissions footprint” is used to describe what we are calculating/estimating, with GWP per passenger as the metric.

3) “The term “non-Kyoto emissions” could be confusing, as it implies a different context. I recommend the authors clarify this terminology or use the more widely accepted “non-CO₂ emissions” instead.”

We appreciate this thoughtful suggestion. However, we have intentionally retained the term “non-Kyoto emissions” to distinguish between greenhouse gases regulated under the Kyoto Protocol and other aviation-specific climate forcers.

While the term “non-CO₂ emissions” is widely used, in the context of aviation it can lead to ambiguity because it technically excludes non-CO₂ gases within the Kyoto protocol (e.g., CH₄ and N₂O). These emissions are already accounted for separately within our LCA framework. Our use of “non-Kyoto emissions” refers specifically to climate forcers not included in the Kyoto Protocol, such as NO_x, H₂O, and contrail-induced cirrus effects.

Given that these are evaluated using global warming potential (GWP) and expressed in CO₂-equivalent (CO₂e) terms we believe “non-Kyoto emissions” offers the most precise and transparent terminology for this context. To reduce any potential for confusion, we have

refined the terminology by saying “non-Kyoto impacts, a term we use to refer to substances created by aeroplanes not covered under the Kyoto Protocol, such as nitrogen oxide (NO_x), water vapor (H₂O), and contrail induced cloudiness (CiC)”. This clarifies that we are not attempting to coin a new term but improve transparency within our study.

- 4) “The literature review is very short, and the gaps in existing research are not sufficiently highlighted. It would strengthen the paper if the authors could provide a more thorough review of relevant literature and more clearly articulate the gaps and their implications for the field. Current relevant papers in this field include:
- Rupcic et al. (2023): Environmental impacts in the civil aviation sector: Current state and guidance. Doi:10.1016/j.trd.2023.103717
 - Cox et al. (2018): Life cycle assessment of air transportation and the Swiss commercial air transport fleet. Doi: 10.1016/j.trd.2017.10.017
 - Rahn et al. (2025): Quantifying climate impacts of flight operations: A discrete-event life cycle assessment approach. Doi: 10.1016/j.trd.2025.104646
 - Keiser et al. (2023): Life cycle assessment in aviation: A systematic literature review of applications, methodological approaches and challenges. Doi: 10.1016/j.jairtraman.2023.102418”

Thank you for these insightful references. They have been included in the manuscript and SI:

- Rupcic et al. reinforces the introduction in the main manuscript in and the goal and scope definition section in SI.
- Cox et al. provides evidence for the importance of upstream emissions, referenced in the Review section in the main manuscript.
- Rahn et al. emphasises the importance of including non-Kyoto impacts and operational scenarios (HAF) into LCA. This is referenced in the literature review and discussion.
- Keiser et al. reinforces the arguments for the importance of a comprehensive LCA-based approach.

- 5) “The statement “The non-Kyoto effects of aviation on climate are substantially greater than the CO₂ effects” in the beginning of chapter 3 is very vague. I recommend providing quantitative data or more specific examples.”

This comment is appreciated, and we have revised the statement to: “The non-Kyoto effects of aviation on climate can be substantially greater than the CO₂ effects” and cited Lee et al.¹.

- 6) “The manuscript emphasizes the necessity of a full cradle-to-grave model to assess the environmental footprint of flights. However, the rationale for this approach is not clear. The authors should clarify what “cradle” and “grave” refer to in the context of aviation and why this model is vital for the assessment.”

We have defined a subsection in the methodology section, now within the main manuscript, to clarify the system scope and the rationale for the full cradle to grave approach in the context of the ATP-DEC.

“Cradle” = Everything from raw-material extraction (mining, refining, composite production) through aircraft assembly and airport/infrastructure construction up to “first flight.”

“Grave” = Everything from aircraft retirement (disassembly, recycling, waste handling) through decommissioning or repurposing of airport facilities.

Why is this vital? Because aviation’s total environmental footprint spans far beyond the fuel uses. A full cradle-to-grave lifecycle model as recommended by ISO 14083 captures hidden CO₂, pollutants, resource depletion, and waste streams that an “in-flight-only” analysis would completely miss and therefore underestimate. By bringing out the life cycle systemic carbon footprint ², actionable offsetting mechanisms ³ can be developed. The cradle-to-grave system scope in aviation sector’s context includes the production and transportation of fuel (Well-To-Tank (WTT)), fuel burning during flying, take-off and landing (Tank-To-Wake (TTW)), in-flight services, and airport and aircraft life cycles. As can be seen in Figure 1, TTW mainly contributed by fuel burning, is a small contributor to the overall GWP. The system consideration is the most comprehensive far beyond the capability of the current aviation sector’s carbon footprint calculator.

7) “The challenges in integrating non-CO₂ effects into assessments are only briefly mentioned. A more thorough literature review on why these effects are difficult to quantify (e.g., due to altitude, location, or other factors) would significantly improve the study's context and depth.”

We have explained the importance of non-Kyoto, its definition, and why it is currently left out of existing tools. This has been addressed in the introduction.

To clarify:

- Non-Kyoto effects are sensitive to atmospheric conditions, altitude, latitude, time of day/season and climate. Also, some impacts have short term climate effects and only persist for hours. For this reason, ATR is preferred as a metric.
- Computationally expensive tools exist that can precisely calculate the contrail and non-CO₂ emissions of historical flights ⁴. However, directly calculating contrail impact for each flight would require complex climate modelling and is unfeasible.
- ATP-DEC address these challenges by using the most advanced available simplified model developed by Dahlmann et al. ⁵. We acknowledge the uncertainty in Dahlmann et al. To help combat one of the uncertainties arising from the use of Dahlmann et al.’s approach, the HAF method considers mean latitude and distance of each flight.
- The non-Kyoto impact is an additional component that can be toggled or updated as science improves. This is another selling point for the modular breakdown design of ATP-DEC.

8) “The content and purpose of figure 1 is not completely clear to me. What is the idea behind this figure? The assumption of an aircraft with only first-class seats is unrealistic and I don’t see the added-value in this graphic.”

This Figure may have been misinterpreted. Figure 1 shows the emissions footprint breakdown for a flight for four existing calculators and ATP-DEC. It includes a comparison of First, Business, Premium Economy, and Economy seats. The figure does not assume an aircraft of only first-class seats. Now this figure may be more clear with the explanation of how the results are produced in manuscript method.

9) “The authors mention the often-overlooked emissions from luggage transportation. However, isn’t this typically included within passenger-related emissions (PAX)? Clarification on this point would strengthen the argument.”

Thank you for your comment and you are correct: luggage transportation is often overlooked. Existing calculators lump a standard luggage allowance into the per-passenger emission calculation. ATP-DEC separates passenger body weight and luggage weight and allocates emissions accordingly. This means in the implementation of ATP-DEC, when a user decides to take more luggage with them (whether its hold luggage or hand luggage), they can add this separately to their weight. This is added value for the ATP-DEC and is now more clearly explained in the methodology found in the main manuscript within the worked example. Table S2 now shows the data behind the luggage factor.

10) “The tools mentioned are difficult for comparison as they do not include non-CO₂ effects. The authors should discuss this limitation more explicitly and explain how their approach provides a more comprehensive assessment.”

Thank you for this comment and we understand the issue. One goal of this article is to encourage change in existing methodologies to include non-Kyoto impacts. We therefore want to make apparent the vast difference between the tools with and without these impacts. This limitation is discussed at length in the comparisons and results.

Existing tools inherently report lower emissions for a given flight in many cases because, as you correctly point out, they often omit non-Kyoto impacts. This is why, for comparison, we have provided a full breakdown of the ATP-DEC output in Figure 1 (this also exemplifies another strength of the ATP-DEC). Readers can understand the impact with and without the non-Kyoto emissions.

Furthermore, when we conduct a direct comparison with the TIM-based model, the non-Kyoto emissions were removed from the ATP-DEC output to allow fair comparison.

11) “The structure of the use cases is unclear. The inclusion of different flights and time frames (e.g., pre-pandemic and pre-Russian occupation periods) lacks organization and makes the methodology hard to follow. A dedicated chapter explaining the methodology and choices made for these use cases/comparisons would greatly enhance clarity.”

The revised SI explains the motivation behind the choice of flight operations data in Section 1.4. To validate the ATP-DEC (specifically the HAF algorithm), historical flight data was acquired commercially through FlightRadar24⁶. To test the HAF algorithm we searched for two flight operation databases either side of a global event, for which we selected the closure of Russian airspace for most airlines in early 2022. The data was therefore taken from 2019 and 2023 to avoid the disruption of the pandemic. Six flight routes were selected because they fly directly over Russia⁷.

12) “The purpose and relevance of the rolling window length (chapter 4.4) are not clearly explained.”

Thank you for your comment. For the HAF, a shorter rolling window might capture recent trends and anomalies but could be noisy, whereas a longer window smooths out fluctuations but might be slower to respond to sudden changes. This section is an investigation and informs the decision of which window to use (by default or by the user) in ATP-DEC. In Section 5.2 we state on revision that 7-day window is adopted as default for the model going forward. This gives the more purpose and relevance to the rolling window length.

13) “What does DOI in table 2 mean?”

Thank you for your comment. This was a misunderstanding on the authors part and has now been corrected to “Accessed: dd mm yy”, meaning the date of access when we conducted the research.

14) “The methodology behind the tool is not sufficiently explained. How does it work and how can a user work apply it? A detailed description of how the tool operates and its underlying assumptions is crucial for understanding and evaluating its validity.”

The revised method section in the main manuscript clarifies how the tool works. We have included a worked example for additional understanding and to link the method with the results.

15) “The manuscript does not allow for reproducibility of the results, which is a fundamental aspect of scientific research. Clear documentation of methods and data sources is needed to address this issue. Additionally, the manuscript does not specify which database version or datasets were used. Including these details is essential for transparency and verification of the results.”

We appreciate the comment and have revised the structure of the manuscript and supplementary information. Following editorial advice and the comments from all peer reviewers, we have moved the fine-tuned methodology to the main manuscript. This helps explain vital aspects of the ATP-DEC. In addition, a step-by-step explanation of an applied example is conducted in the SI, with cross referencing to improve readability and interconnectivity between the methodology and results.

In terms of documentation of data sources, we have ensured that all data sources are referenced in the manuscript and SI.

16) “While the manuscript claims to consider “full life flight operations,” it does not follow the standardized LCA methodology (based on ISO 14040/44). Key elements such as the definition of goals, scope, and inventory creation are unclear, which undermines the transparency of the study.”

We appreciate the comment. The main manuscript now contains the revised methodology with the structure more clearly following the standardised ISO 14040/44 LCA methodology. This is achieved by a designated Goal and Scope Definition section and the basis for Inventory Analysis and Impact Assessment. The former clarifies the system boundary and scopes as well as objectives of the study. The latter is equation or formulation driven and has been detailed through the equations for each life cycle activity. The interpretation shows the step-by-step calculations to use ATP-DEC and generate the results, for example, as those illustrated in the main paper. The transparency has been improved through illustrative examples for each formulation. This can be found in the method in the main manuscript, where the application example from Figure 1 is explained step by step, each equation substituted with the parametric and variable values.

17) “The inclusion of non-CO₂ effects is presented as a unique selling point. However, the equations used appear to be entirely adapted from Dahlmann et al. (2023), where specific assumptions for their validity are explicitly stated. It is unclear whether these assumptions are actually met in the context of this manuscript. The authors should carefully evaluate whether the assumptions underlying these equations are satisfied and clearly justify their applicability to this study”

In section 5.4 an additional model limitation has been noted about the Dahlmann et al method and its assumptions. The estimated mean error of the method for calculating non-Kyoto emissions has now been noted and compared to the estimated mean error of applying a constant multiplier for non-Kyoto emissions.

18) “The supplementary material is very detailed, organized, and useful, but it is difficult to fully understand the paper without any explanation of the aspects presented in it. It would be helpful to clarify how and why these elements are relevant to the study within the main text, as this would improve the overall readability and comprehension of the paper.”

We appreciate the comment and have revised the structure of the manuscript and supplementary information. Following editorial advice and the comments from all peer reviewers, we have moved the fine-tuned methodology to the main manuscript. This helps explain vital aspects of the ATP-DEC. In addition, a step-by-step explanation of an applied example is conducted in the SI, with cross referencing to improve readability and interconnectivity between the methodology and results.

Reviewer 3 Summary

19) “Provide a more detailed explanation of the methodology and the use cases for a better overall understanding.”

Thank you for your detailed comments on the methodology. We believe we have understood your viewpoint and improved the explanation of the methodology substantially in the manuscript and SI. The manuscript now clearly explains the step-by-step method with the outline of the example calculation. The SI now goes through the example in detail. Throughout the text the figures and equations are referenced for clarity.

20) “Clarify the use and impact of historical adjustment factors, and consider including more direct comparisons with other models.”

Thank you for taking the time to understand the HAF. We have clarified the use and impact of HAF through a detailed explanation of the example calculation with an example equation. Figures 1, 2, 3, 4, and 5 all show the impact of the HAF and Figures 1 and 2 show direct comparison with other models.

21) “Ensure greater transparency regarding the databases, inventories, and models (e.g.,ecoinvent), and clearly state the contributions made by the authors beyond existing literature.”

We appreciate the comments on transparency, since this is one of the main goals of the ATP-DEC: full transparency. We have referenced all databases used, in both the main manuscript and the SI. All data used in the results has been referenced and some has been added in tables in the SI for full reproducibility.

Each author has made a substantial contribution to the concept and design of the work. The contributions of the authors include (CRediT Roles):

F McFall – Methodology, Software, Validation, Investigation, Data Curation, Writing – Original Draft, Visualisation, Formal Analysis

D Guan – Methodology, Resources, Writing – Review & Editing

J Chenoweth – Methodology, Validation, Investigation, Writing – Original Draft, Supervision

X Font – Methodology, Validation, Investigation, Writing – Original Draft, Supervision

I Corduneanu – Conceptualisation, Investigation, Methodology, Resources, Writing – Review & Editing, Project Administration, Funding Acquisition

E Goean – Conceptualisation, Methodology, Resources, Writing – Review & Editing, Supervision, Project Administration, Funding Acquisition

J Sadhukhan – Methodology, Validation, Investigation, Writing – Original Draft, Supervision

References

- 1 Lee, D. S. *et al.* The contribution of global aviation to anthropogenic climate forcing for 2000 to 2018. *Atmospheric environment* **244**, 117834 (2021).
- 2 Sadhukhan, J. Net zero electricity systems in global economies by life cycle assessment (LCA) considering ecosystem, health, monetization, and soil CO₂ sequestration impacts. *Renewable Energy* **184**, 960-974 (2022).
- 3 Goean, E. R. *et al.* Using the Blockchain to Reduce Carbon Emissions in the Visitor Economy. *Sustainability* **16**, 4000 (2024).
- 4 Estuaire. Estuaire: Climate analytics for aviation. *Estuaire* (2024).
- 5 Dahlmann, K., Grewe, V., Matthes, S. & Yamashita, H. Climate assessment of single flights: Deduction of route specific equivalent CO₂ emissions. *International Journal of Sustainable Transportation* **17**, 29-40 (2023).
- 6 FlightRadar, A. B. FlightRadar24: Live Flight Tracker. *FlightRadar24* (2025).
- 7 Chu, C., Zhang, H., Zhang, J., Cong, L. & Lu, F. Assessing impacts of the Russia-Ukraine conflict on global air transportation: From the view of mass flight trajectories. *Journal of Air Transport Management* **115**, 102522 (2024).